# Discovering Factor Level Preferences to Improve Human-Model Alignment

## Abstract

Despite advancements in Large Language Model (LLM) alignment, understanding the reasons behind LLM preferences remains crucial for bridging the gap between desired and actual behavior. LLMs often exhibit biases or tendencies that diverge from human preferences, such as favoring certain writing styles or producing overly verbose outputs. However, current methods for evaluating preference alignment often lack explainability, relying on coarse-grained comparisons. To address this, we introduce PROFILE (PRObing Factors of InfLuence for Explainability), a novel framework that uncovers and quantifies the influence of specific factors driving preferences. PROFILE's factor level analysis explains the "why" behind human-model alignment and misalignment, offering insights into the direction of model improvement. We apply PROFILE to analyze human and LLM preferences across three tasks: summarization, helpful response generation, and document-based question-answering. Our factor level analysis reveals a substantial discrepancy between human and LLM preferences in generation tasks, whereas LLMs show strong alignment with human preferences in evaluation tasks. We demonstrate how leveraging factor level insights, including addressing misaligned factors or exploiting the generation-evaluation gap, can improve alignment with human preferences. This work underscores the importance of explainable preference analysis and highlights PROFILE's potential to provide valuable training signals, driving further improvements in human-LLM alignment.

## 1 Introduction

Large Language Models (LLMs) are widely recognized for their ability to generate human-level texts, yet they often fail to fully align with human preferences. Despite significant advancements in alignment techniques like Reinforcement Learning from Human Feedback (RLHF) (Ouyang et al., 2022) and Direct Preference Optimization (DPO) (Rafailov et al., 2024), LLMs tend to exhibit biases toward specific writing styles (Das et al., 2024) or generate overly verbose outputs (Park et al., 2024). Understanding the underlying factors contributing to this misalignment is essential for enhancing LLM performance.

Previous attempts to understand and improve preference alignment (Ouyang et al., 2022; Rafailov et al., 2024; Song et al., 2024) have primarily relied on coarse-grained approaches, lacking explainability. These methods often focus on identifying which model is preferred overall but do not provide insights into the factors that drive these preferences. While some studies analyze human preferences at a finer granularity (Hu et al., 2023; Kirk et al., 2024; Scheurer et al., 2023), a comparative analysis of how these preferences align with models remains limited. Furthermore, existing evaluation approaches often suffer from limited scalability and generalizability across diverse tasks and settings due to their heavy reliance on human annotation (Chiang et al., 2024; Zheng et al., 2023).

To address these limitations in explainability and generalizability, we introduce PROFILE (PRObing Factors of InfLuence for Explainability), a novel analytical framework designed to uncover and quantify the key factors driving both human and model preferences. Our framework analyzes pairwise preference data to measure how specific factors manifest in preferred responses, enabling us to rank the relative influence of different factors and compare these rankings between humans and

---

*Equal contribution.

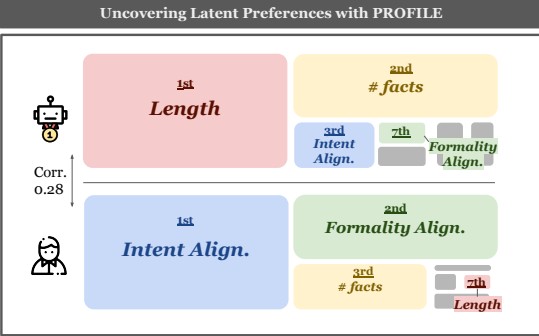

| Model Behavior & Human Reaction | Uncovering Latent Preferences with PROFILE |

Summarize this Post: ...

This is the summary of the Post:
A college freshman, new to Reddit, is seeking inspiration and ideas for an observational study on Internet censorship for their English 102 class. (...) They apologize for the lengthy post and request assistance from the Reddit community.

*Hmm... My best model from the leaderboard generates better responses than other models but I'm **not totally satisfied!***

We **cannot clearly pinpoint** how human preferences and model generations misalign.

We **quantify each factor's influence** and **measure correlations** between human and models.

Figure 1: LLMs often fall short of expectations, but pinpointing *why* is challenging (left panel). PROFILE quantifies the influence of various factors on preferences, revealing the underlying causes of human-model misalignment (right panel).

models (Figure 1). PROFILE offers a more granular, factor-level understanding, providing actionable insights for improving LLM alignment. Furthermore, it is applicable across various tasks and settings, enabling comprehensive analysis of model behavior both as a text generator and as an evaluator. This dual-purpose analysis is particularly crucial as LLMs increasingly serve as evaluators that assess and provide feedback on text quality for AI training (Bai et al., 2022; Lee et al., 2023; Guo et al., 2024).

Using PROFILE, we investigate three key research questions: **RQ1.** How well do LLM-generated responses align with human preferences at a factor level? **RQ2.** How well do LLMs' judgments align with human preferences at a factor level when evaluating responses? **RQ3.** Can we leverage insights gained from factor level analyses to enhance LLM alignment?

To answer these questions, we analyze model preferences in both generation and evaluation settings across three tasks: summarization, helpful response generation, and document-based QA, commonly used for preference optimization. We compare the preferences of eight LLMs, including open-source and proprietary models, against human preferences at a granular factor level. Our results reveal a significant discrepancy in generation settings, with the best-aligned model achieving only a 0.289 correlation with human preferences. Notably, LLMs consistently prioritize length across all tasks, contrary to human preferences. However, in evaluation settings, LLMs show a surprising alignment with human judgments, with the best model reaching a 0.822 correlation with humans.

Leveraging these insights, we show that factor level analysis can significantly improve LLM alignment. In the summarization task, we find that prompting LLM evaluators with guidance on misaligned factors identified by PROFILE improves the overall evaluation accuracy. Using feedback from LLMs as evaluators, which exhibit closer alignment to human preferences than LLMs as generators, improves the factor level alignment of model-generated output. These findings suggest PROFILE can provide valuable training signals for improving human-LLM alignment.

Our contributions are as follows:

- We present PROFILE, a framework for analyzing factor level preferences in human-LLM alignment. PROFILE is adaptable across tasks, operates without fine-grained human annotations, and enables scalable analysis of both human and LLM in various settings.

- Using PROFILE, we identify significant misalignments between human and LLM preferences in text generation, revealing that LLMs prioritize certain factors differently from humans, even when their overall performance appears strong. Notably, we show that LLMs align more closely with human preferences in evaluation than in generation setting.

- We show that the factor level understanding from PROFILE's explainable analysis in both generation and evaluation settings, along with the insights from comparing these settings, can help improve human-LLM alignment.

## 2 PROBLEM DEFINITION

To address our central question of how well LLMs align with human preferences, we acknowledge the multifaceted nature of human preference where a perceived quality of response depends on various factors. To uncover these latent preferences, we define a set of factors $\mathcal{F} = \{f_1, f_2, \ldots, f_n\}$, which influence human preferences. Each $f_i$ represents a specific aspect of the text (e.g., fluency, length). We then quantify their influence on human preference as *factor-level preferences*, denoted by $\mathbf{f}(\mathcal{R})$.

$$\mathbf{f}(\mathcal{R}) = (f_1(\mathcal{R}), f_2(\mathcal{R}), \ldots, f_n(\mathcal{R})),$$

where $f_i(\mathcal{R})$ represents the influence of each factor ($f_i$) on the overall preference across the response set $\mathcal{R}$. We refer to $f_i(\mathcal{R})$ as the ***factor score*** of factor $f_i$. We extend this concept to include both humans and models, referring to both as "agents."

### 2.1 OPERATIONAL DEFINITIONS

We determine factor-level preferences $\mathbf{f}(\mathcal{R})$ by analyzing observable response-level preferences in a pairwise comparison setting. This setting refers to a scenario where an agent compares two responses, e.g. $r_i$ and $r_j$, and selects the more preferred one (either $r_i$, $r_j$, or a tie). The operational definitions of the pairwise preferences required for this experiment are defined as follows.

**Pairwise Preferences.** We define the pairwise preference function for a pair of two responses as:

$$Pref : \mathcal{R} \times \mathcal{R} \rightarrow \{-1, 0, 1\}$$

where $Pref(r_i, r_j) = 1$ if response $r_i$ is preferred over response $r_j$; $Pref(r_i, r_j) = -1$ if response $r_j$ is preferred over response $r_i$; and $Pref(r_i, r_j) = 0$ if the preference between $r_i$ and $r_j$ is a tie. In our experiments, we define model pairwise preferences for both generation and evaluation settings.

**Model Pairwise Preferences in Generation.** While models might not have preferences in the same way humans do, we can operationally define the preferences of a model through the responses it generates at different score levels. Specifically, if a model assigns scores of 3 and 5 to two responses, the response with a score of 5 is considered "preferred" by the model over the response with a score of 3. To implement this systematically, we prompt the model to generate responses corresponding to predefined scores ranging from 1 to 5, forming the set $\mathcal{R} = \{r_{\text{score}} \mid \text{score} \in \{1, 2, 3, 4, 5\}\}$. Pairwise Preferences in Generation, $Pref_{gen}$, is defined by comparing the model-assigned scores $Score(r_i)$ and $Score(r_j)$. Specifically, $Pref_{gen}(r_i, r_j) = 1$ if $Score(r_i) > Score(r_j)$ and $Pref_{gen}(r_i, r_j) = -1$ if $Score(r_i) < Score(r_j)$. This approach is inspired by methods used in constructing training data for evaluator models (Kim et al., 2023).

**Model Pairwise Preferences in Evaluation.** We define model preferences in an evaluation setting, similar to the general approach used to measure human preferences. Given two responses $r_i$ and $r_j$, the model selects which is the better response. Pairwise Preferences in Evaluation, $Pref_{eval}(r_i, r_j) = 1$ if the model evaluates $r_i$ as preferable over $r_j$; $Pref_{eval}(r_i, r_j) = -1$ if $r_j$ is preferred over $r_i$; and $Pref_{eval}(r_i, r_j) = 0$ if the model finds them equally preferable (tie). This approach, where models make pairwise preference evaluation, is similar to how LLMs generate preference labels (Lee et al., 2023). Although we extract model preferences separately for evaluation and generation tasks, we assume that human preferences remain consistent across both, as human judgments are always based on evaluating model-generated outputs.

**Pairwise Factor Comparison.** For each factor $f_k$, we define a function $M_k$ to compare factor's manifestation in pairs of responses:

$$M_k : \mathcal{R} \times \mathcal{R} \rightarrow \{-1, 0, 1\}$$

Specifically, $M_k(r_i, r_j) = 1$ if factor $f_k$ is more manifest in response $r_i$; $M_k(r_i, r_j) = -1$ if factor $f_k$ is more manifest in response $r_j$; and $M_k(r_i, r_j) = 0$ if factor $f_k$ is equally manifest in both responses. For example, if $f_k$ represents length and $r_i$ is longer than $r_j$, $M_{length}(r_i, r_j) = 1$.

## 3 PROFILE: PRObing Factors of InfLuence for Explainability

We introduce PROFILE, a novel method for automatically quantifying the influence of specific factors on both human and model preferences, revealing *factor-level preferences*. Building on the

work of Hu et al. (2023), which analyzes factors influencing human preferences, PROFILE extends this analysis to assess preference alignment between humans and models by identifying the driving factors behind these preferences.

We first establish a comprehensive taxonomy of fine-grained factors to guide the selection of appropriate factor sets $\mathcal{F}$ for the tasks (§ 3.1). We then detail methods for quantifying the influence of each factor, $f_i(\mathcal{R})$, enabling us to determine *factor-level preferences* for each agent and analyze their alignment (§ 3.2). PROFILE's versatility across various agents, tasks, and settings (generation and evaluation) makes it a powerful tool for comprehensive preference alignment analysis.

## 3.1 TAXONOMY DESIGN

| Level 3 | Level 2 | Level 1 | Definition |
|---|---|---|---|
| Input-Output | Relevance | Receptiveness | Whether the core question of the input has been answered. |
| | | Off Focus | The ratio of atomic facts that are not related to the main focus of the input. |
| Source-Output | Consistency | Intent Align. | Whether the intent of the source and output is the same. |
| | | Hallucination | The ratio of atomic facts that are incorrect compared to the original source. |
| | | Source Coverage | The ratio of atomic facts in the source that appear in the output. |
| | Linguistic Style | Formality Align. | Whether the formality of the source and output is the same. |
| | | Novel Words | The ratio of words in the output that are not used in the source. |
| Output-Only | | Length | The number of words used in the output. |
| | | Fluency | The quality of individual sentences. |
| | Informativeness | Number Of Facts | The number of atomic facts in the output. |
| | | Helpfulness | The ratio of facts that provide additional helpful information. |
| | Safety | Misinformation | The ratio of facts that include potentially incorrect or misleading information. |
| Intra-Output | Coherence | Coherence | Whether all the sentences form a coherent body. |

Figure 2: The full taxonomy and definitions of Level 1 factors.

We introduce a comprehensive taxonomy of fine-grained factor for evaluating preference alignment between human and model in diverse set of text generation tasks. Addressing the lack of a unified framework and inconsistent terminology in existing literature, we incorporate evaluation factors from various tasks, including summarization, helpful response generation, question answering, and instruction following (Zhong et al., 2022; Fabbri et al., 2021; Hu et al., 2023; Fu et al., 2024; Ye et al., 2024; Glaese et al., 2022; Nakano et al., 2021).

Our three-level taxonomy comprises: (i) **Level 1**: 13 distinct factors directly related to preference alignment; (ii) **Level 2**: Groups of related Level 1 factors based on shared characteristics (e.g., Length and Fluency fall under "Linguistic Style"); and (iii) **Level 3**: Categories defined by the relationship each factor examines: Input-Output (relationship between input and output), Source-Output (relationship between source text and output), Output-Only (characteristics of the output itself), and Intra-Output (relationship among sentences within the output). Levels 1 and 2 are derived from existing studies, while Level 3 is designed to provide a structured perspective on factor relationships. The complete taxonomy is detailed in Figure 2.

This hierarchical structure guides factor selection based on the task. For example, source-dependent tasks (e.g., summarization) require factors from all three high-level categories, while input-driven tasks (e.g., QA) focus on Input-Output and Intra-Output factors.

## 3.2 QUANTIFICATION OF HUMAN-MODEL PREFERENCE ALIGNMENT

This section outlines the process of quantifying *factor-level preferences* and measuring the alignment of these preferences between humans and the model. First, we calculate *factor score* $f_i(\mathcal{R})$ by comparing the pairwise preference ($Pref$) with the factor-specific pairwise comparison ($M_k$) across the set of all possible response pairs in the dataset. These scores are then used to rank the factors, and the alignment between human and model preferences at the factor level is quantitatively evaluated based on these rankings.

**Automatically Determining Factor Manifestation** To analyze the manifestation of our factors in model and human-preferred responses and determine $M_k$, we develop an automatic factor extraction framework. We employ three approaches based on the objectivity of each factor: (i) Rule-based:

For straightforward, objective factors, we use deterministic algorithms. Length and Novel Words are extracted this way. (ii) UniEval-based: For inherently subjective factors (Fluency and Coherence), we use the well-established UniEval metric (Zhong et al., 2022). UniEval is a learned metric that provides scores of range 0-1 for various aspects of text quality. (iii) LLM-based: For factors that rely on objective criteria but require more nuanced judgment, we use GPT-4o with carefully designed prompts. This approach is further divided into "response-based" (Intent Alignment and Formality Alignment) and "atomic-fact-based" (the remaining seven) extraction depending on the level of detail needed for each factor. By combining these three approaches, our framework captures a wide range of factors with appropriate levels of objectivity. The specific details of the implementation of each method and validation of LLM-based extractions can be found in Appendix D.

**Quantifying Influence of Each Factor.** To quantify the influence of each factor, i.e., *factor score*, we use $\tau_{14}$, a variation of Kendall's correlation proposed by Deutsch et al. (2023). This metric is well-suited for handling the distribution of ties, particularly in our setting, where ties arise in only one of the comparison sets used for calculating Kendall's $\tau$. Below, we explain the specific ways ties appear in our analysis.

Since our analysis relies on pairwise comparisons, we calculate $\tau_{14}$ for each factor $f_k$ using pairwise concordance and discordance, following the methodology outlined by Bojar et al. (2017). The metric is defined as:

$$\tau_{14}(f_k) = \frac{|C_k| - |D_k|}{|C_k| + |D_k| + |T_k|},$$

where $C_k$ is the count of concordant pairs, where the overall preference and the manifestation of factor $f_k$ agree, $D_k$ is the count of discordant pairs, where the overall preference and the manifestation of factor $f_k$ disagree, and $T_k$ is the count of ties, are handled differently depending on the context. Mathematically, $C_k$ and $D_k$ are computed as:

$$C_k = \sum_{r_i, r_j \in R, i < j} \mathbb{1}[Pref(r_i, r_j) \cdot M_k(r_i, r_j) = +1],$$

$$D_k = \sum_{r_i, r_j \in R, i < j} \mathbb{1}[Pref(r_i, r_j) \cdot M_k(r_i, r_j) = -1],$$

where $\mathbb{1}[\text{condition}]$ is 1 if the condition is true and 0 otherwise.

In our experimental setup, the definition of $T_k$ depends on the specific setting. (1) In the **generation** setting, no ties exist in response preferences because models do not generate responses with identical scores. Therefore, $T_k$ is defined as the occurrence of ties at the factor level, which is calculated as the number of instances where $M_k(r_i, r_j) = 0$. (2) In the **evaluation** setting, ties at the factor level (e.g., pairs with the same length) are removed to allow for a clearer analysis of the factor's influence. In this case, $T_k$ is the number of occurrences where $(\text{Pref}(r_i, r_j) = 0)$.

For instance, consider the factor $M_{length}$, which measures response length. If response $r_1$ is longer than $r_2$ ($M_{length}(r_1, r_2) = 1$) and the model prefers $r_1$ ($Pref(r_1, r_2) = 1$), this pair is classified as concordant. Conversely, if the model prefers the shorter $r_1$, the pair is discordant. Evaluating all pairs, a positive factor score indicates a positive influence of the factor, a negative score indicates a negative influence, and a score close to zero implies minimal influence. The magnitude of the score reflects the strength of this influence.

**Evaluating Factor-Level Preference Alignment.** An agent's *factor-level preferences* are defined as a ranking of factors based on their scores, where a higher rank and score indicate a stronger influence of that factor on the agent's overall preference. The correlation between human and model rankings reflects their agreement on the relative importance of factors to overall preference, which we use as a measure of factor-level preference alignment between humans and models. We calculate Spearman's $\rho$, Kendall's $\tau$ [*], and Pearson's $r$ coefficients to quantify this alignment.

## 4 ANALYZING PREFERENCE ALIGNMENT THROUGH PROFILE

This section details the experimental setup used to address our research questions (§ 4.1). Results for RQ1, RQ2, and RQ3 are presented in Sections § 4.2, § 4.3, and § 4.4, respectively.

---

[*] We use Kendall's $\tau_b$ (Kendall, 1945) as the default.

## 4.1 EXPERIMENTAL SETTING

**Tasks and Models.** We analyze three publicly available datasets used in preference optimization methods: (i) Reddit TL;DR (Stiennon et al., 2020), which includes human ratings of summaries across multiple evaluation dimensions; (ii) StanfordHumanPreference-2 (SHP-2) (Ethayarajh et al., 2022), focusing on human preferences over responses in the "`reddit/askacademia`" domain; and (iii) OpenAI WebGPT (Nakano et al., 2021), which compares model-generated answers on the ELI5 subreddit based on factual accuracy and usefulness. We refer to the tasks for each dataset as summarization, helpful response generation, and document-based QA tasks in this paper. We exclude pairs with human Tie ratings in all three datasets, as our analysis focuses on cases with clear preference distinctions. For our experiments, we utilize both open-source and proprietary LLMs. Open-source models include LLaMA 3.1 70B (Dubey et al., 2024), Mixtral 8x7B Instruct v0.1 (Jiang et al., 2024), and three TÜLU v2.5 models (Ivison et al., 2024) (TÜLU v2.5 + PPO 13B (13B RM), TÜLU v2.5 + PPO 13B (70B RM), and TÜLU v2.5 + DPO 13B). Proprietary models include Gemini 1.5 Flash (Reid et al., 2024), GPT-4o (OpenAI, 2024), and GPT-3.5. From here on, we refer to Gemini 1.5 Flash as Gemini 1.5, Mixtral 8x7B Instruct v0.1 as Mixtral, TÜLU v2.5 models as Tulu 2.5 + {alignment training strategy}. Detailed descriptions of the datasets and models can be found in Appendix C.2.

**Experimental Setup.** For each task, we explore two settings: (i) Generation, where models generate responses that would receive a score of 1-5 for a given task, and (ii) Evaluation, where models select the better of two provided responses, which are taken from the datasets. See Appendix E for prompts. In both settings, we use PROFILE to extract factor scores and their factor rankings and measure the correlation with human judgments (factor-level preference alignment). In addition to factor-level analysis, we assess overall pairwise response agreement between humans and models. For evaluation, we report the percentage of models' agreement with existing human labels by measuring how often it aligns with human judges' selections of the better response.

## 4.2 ARE MODELS ALIGNED WITH HUMAN PREFERENCE AT A FACTOR-LEVEL IN GENERATION TASKS?

Human and model preferences consistently misalign at the factor level across summarization, helpful response generation, and document-based QA (Figure 3). Models consistently prioritize Length across all tasks (right-hand side of the figure), while human priorities vary. In the summarization task (Figure 3a), humans prioritize Intent Alignment (0.596) and Formality Alignment (0.594), while models focus on Length (GPT-4o: 0.978, Gemini 1.5: 0.906), often generating longer summaries for higher scores. Notably, humans dislike summaries with many new words (factor score -0.167 for Novel Words), yet models produce more novel words in high-scoring outputs (GPT-4o: 0.472, Gemini 1.5: 0.56). The numbers in parentheses represent factor scores. In the helpful response generation task (Figure 3b), humans prioritize Receptiveness and Helpfulness, but their overall factor scores are relatively low (0.248, 0.193 respectively), indicating no single dominant factor drives their preferences in this task. In contrast, models exhibit much stronger preferences, again emphasizing Length and Number Of Facts. For document-based QA (Figure 3c), humans prioritize Receptiveness and prefer answers without Hallucinations, aligning with the need for factual accuracy of the task. However, models still heavily emphasize Length (0.965 for both GPT-4o and Gemini 1.5) and also prioritize Coherence and Helpfulness more than humans do.

This misalignment is quantified by low *factor-level preference alignment* ($\tau$). The left Generation column in Table 1 shows that even the best-performing model (Gemini 1.5) only achieves a 0.289 $\tau$ correlation with human preferences in summarization task. Similar low correlations are observed in other tasks (Appendix, Table 9). Full factor scores are available in Appendix Table 8. A small-scale annotation exploring human evaluation of model-scored responses, including an example of disagreement, is presented in Appendix A.

## 4.3 ARE MODELS ALIGNED WITH HUMAN PREFERENCES AT A FACTOR-LEVEL IN EVALUATION TASKS?

Our analysis reveals a consistent trend of stronger alignment between models and human preferences in evaluation tasks compared to generation tasks. Table 1 demonstrates this by showing *factor-level preference alignment* of human and model, measured using Kendall $\tau$, Spearman $\rho$, and Pearson $r$

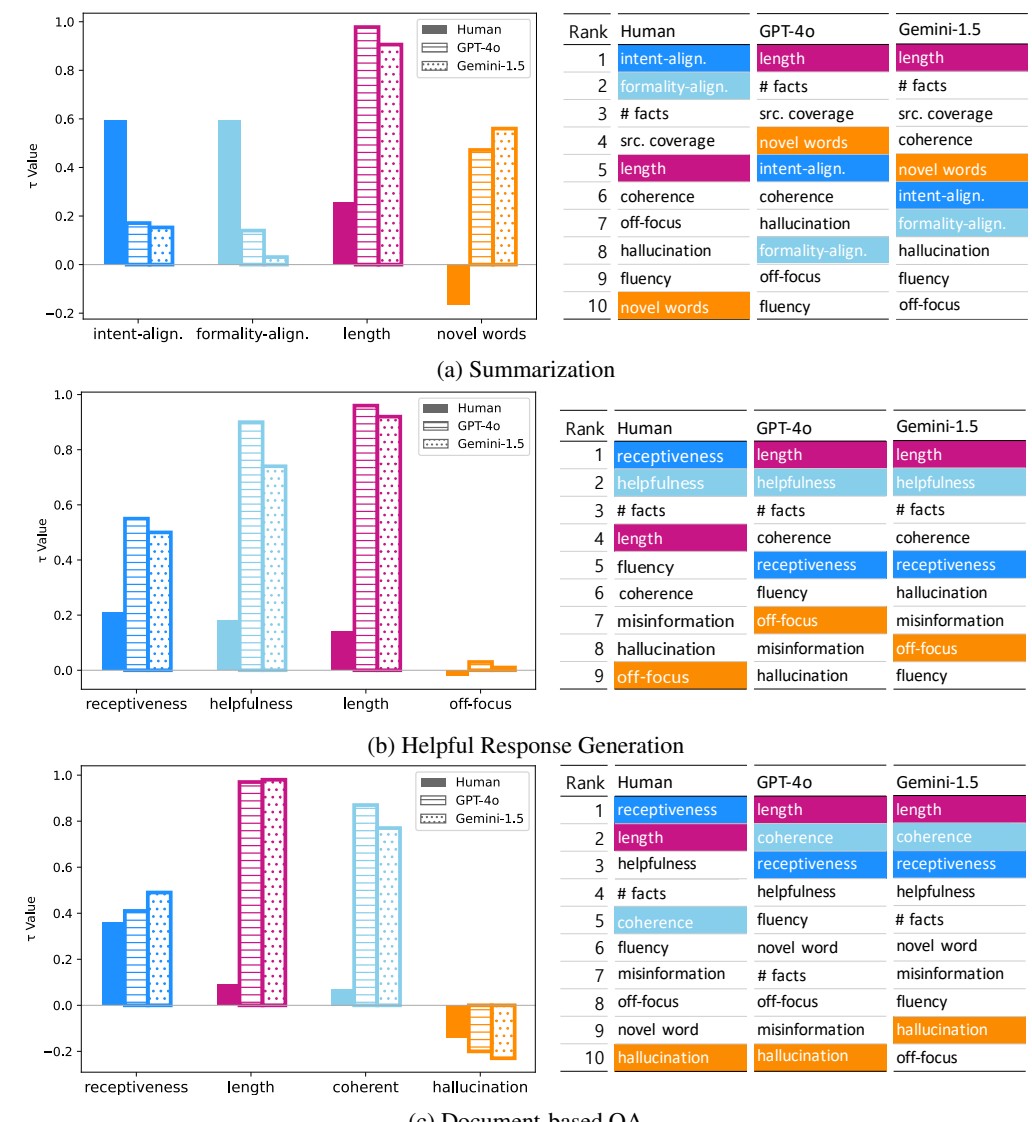

Figure 3: Comparison of factor-level preference alignment between humans, GPT-4o, and Gemini-1.5 in generation across three tasks: (a) Summarization, (b) Helpful Response Generation, and (c) Document-based QA. The left bar graphs display *factor scores* ($\tau_{14}$) for selected factors. The right tables show the rankings of all factors for each task. Notably, both models consistently rank 'length' as the top factor across tasks, while human preferences vary by task.

correlations, are consistently higher in the evaluation setting across all models. For instance, GPT-4o exhibits the highest alignment in evaluation ($\tau$: 0.822, $\rho$: 0.939, $r$: 0.983) but much lower alignment in generation ($\tau$: 0.156, $\rho$: 0.297, $r$: 0.155).

The observed disparity between generation and evaluation performance resonates with the emerging understanding of the paradoxical behaviors of generative AI models (West et al., 2023; Oh et al., 2024). Despite both tasks being fundamentally next-token prediction tasks, factor-level preference alignment with humans differs significantly. This gap is further highlighted in our analyses of GPT-4o-generated feedback (§ 4.4)), where GPT-4o accurately critiques aspects of its own generated summaries (e.g., "unnecessary specifics (like the exact ages and the name of the allergy site)") that contradict its priorities in generations (e.g., Source Coverage and Number Of Facts). This disparity between evaluation and generation performance motivates us to explore the potential for utilizing the differences in evaluation and generation performance to improve alignment in generation.

| | Generation | | | Evaluation | | | |
|---|---|---|---|---|---|---|---|
| | $\tau$ | $\rho$ | $r$ | $\tau$ | $\rho$ | $r$ | *Agree. (%)* |
| Mixtral | 0.200 | 0.297 | 0.069 | 0.244 | 0.382 | 0.453 | 0.526 |
| Tulu 2.5 + PPO (13B RM) | -0.156 | -0.164 | -0.189 | 0.511 | 0.685 | 0.739 | 0.516 |
| Tulu 2.5 + PPO (70B RM) | 0.111 | 0.200 | -0.015 | 0.644 | 0.830 | 0.844 | 0.520 |
| LLaMA 3.1 70B | 0.111 | 0.248 | 0.213 | 0.733 | 0.903 | 0.975 | 0.705 |
| Gemini 1.5 | **0.289** | **0.394** | **0.171** | 0.778 | 0.915 | 0.972 | 0.721 |
| GPT-4o | 0.156 | 0.297 | 0.155 | **0.822** | **0.939** | **0.983** | **0.784** |

Table 1: Factor-level preference alignment($\tau$, $\rho$, $r$) between model and human in generation and evaluation settings, and overall evaluation agreement rate for Summarization task. For Tulu PPO models, the size in the parentheses is the size of the RM used to train the LLMs.

For some models, despite similar overall pairwise preference agreement rates, factor-level preference alignment differs significantly. This is evident in the comparison of Tulu 2.5 + PPO (13B RM) ($\tau$: 0.511) and Mixtral ($\tau$: 0.244), which have comparable overall agreement rates (0.516 and 0.524, respectively). Our factor-level analysis reveals subtleties in model alignment that overall agreement rates fail to capture. A qualitative examination of factor scores and their rankings (Table 6 in the Appendix G) reveals that, despite both models rank near the bottom in overall agreement in evaluation, Tulu 2.5 + PPO (13B RM) exhibits a stronger correlation with human factor rankings and demonstrates a more significant influence of those factors. Additionally, we analyze the correlations between features for each model, and the correlation matrices can be found in the Appendix.

## 4.4 ACHEIVING BETTER ALIGNMENTS THROUGH PROFILE

**Improving Alignment in Evaluation through Factor-level Guidance.** One of the key features of our approach is its explainability of human-LLM misalignment. To evaluate whether insights from PROFILE can enhance model performance, we conduct an experiment using a summarization task with Mixtral and Tulu 2.5 + PPO (13B RM), providing LLM evaluators with factor-specific guidance. Two strategies are used in the prompts: $\text{Guide}_{Rand}$ (guidance on a randomly selected factor) and $\text{Guide}_{Mis}$ (guidance on a factor where model and human preferences significantly diverge). The guidance explicitly mentions the target factor and its definition. See Appendix F.1 for experiment details including the specific factors and prompts.

Across 200 response pairs for each model, $\text{Guide}_{Mis}$ yields a significant increase in evaluation agreement with humans compared to both $\text{Guide}_{Rand}$ and the baseline agreement (without any guidance, calculated on the same 200 pairs). These results, presented in Table 2, strongly suggest that tailoring guidance to address specific misalignments effectively im-

| | Base. | $\text{Guide}_{Rand}$ | $\text{Guide}_{Mis}$ |
|---|---|---|---|
| Tulu 2.5 | 0.529 | 0.532 | **0.578** |
| Mixtral | 0.651 | 0.644 | **0.664** |

Table 2: Evaluation Agreement(%) on Baseline and $\text{Guide}_{Rand}$, and $\text{Guide}_{Mis}$ settings.

proves model performance and alignment with human expectations, highlighting the value of our factor-level analysis.

**Leveraging Evaluation for Better Alignment in Generation.** Prior analysis shows that models have stronger factor-level alignment during evaluation than generation (Section 4.3), suggesting that evaluator feedback might improve generation alignment. To test this, we conduct an experiment on feedback-driven summary improvement: a generator model produces two initial summaries per input, and an evaluator model selects the preferred summary (or tie) and its justification. The generator then uses this feedback to create an improved summary.

We compare this with two baselines: (1) $\text{Baseline}_A$, where the generator produces one improved summary from both initial summaries *without* feedback; and (2) $\text{Baseline}_B$, where it generates two improved summaries *without* feedback, each based on one initial summary. This simulates a common generation improvement scenario where improvement relies on an implicit critique of a single text piece. The experiment uses 100 Reddit TL;DR samples with three generators (GPT-4o, LLaMA 3.1 70B, and Tulu 2.5 + PPO (70B RM)) and the top-performing evaluator (GPT-4o.).

|  | GPT-4o | | LLaMA 3.1 70B | | Tulu 2.5 + PPO (70B RM) | |
|---|---|---|---|---|---|---|
|  | $\tau_G$ | $\tau_H$ | $\tau_G$ | $\tau_H$ | $\tau_G$ | $\tau_H$ |
| Baseline$_A$ | -0.24 | $-0.07$ | $-0.20$ | $-0.29$ | $-0.29$ | $-0.29$ |
| Baseline$_B$ | -0.29 | $-0.29$ | $-0.42$ | $-0.42$ | $-0.24$ | $-0.24$ |
| GPT-4o feedback | **0.36** | **0.45** | **0.29** | **0.20** | **0.16** | **0.16** |

Table 3: Factor-level alignment ($\tau$) between improvements made by different generators (GPT-4o, LLaMA 3.1 70B, Tulu 2.5 + PPO (70B RM)) and factor-level preferences from GPT-4o (evaluation) and human. $\tau_G$ indicates the degree of alignment with GPT-4o preferences, while $\tau_H$ indicates alignment with human preferences. Higher values signify a stronger alignment of improvements with the factor-level preferences of human or GPT-4o evaluators.

Table 3 illustrates that for all three generators, incorporating evaluator feedback during the improvement process leads to a positive change, correlating with both GPT-4o and human judgments. In contrast, both baselines exhibit negative correlations, indicating a divergence from the desired preferences. These findings emphasize that leveraging external evaluation feedback, rather than relying solely on self-improvement, is more effective for enhancing alignment in text generation. Manual analysis of 30 samples confirms that higher-ranked factors in the evaluator's factor-level preferences are more prominent in the evaluator's feedback, except for Formality Alignment (see Appendix F.2.3). Details of the prompts used and the metrics can be found in Appendix F.2.1-F.2.2.

## 5 DISCUSSION

**Alignment of Reward Models and Language Models.**

To understand whether preference misalignment originates from reward models (RMs), we compare factor-level alignment between RM, their corresponding RLHF-trained LLM, and human preferences in a summarization task.

Figure 4 shows the factor-level alignment ($\tau$) between human preferences and those of RMs and LLMs in both generation and evaluation settings. The results indicate that RMs have a stronger alignment with human preferences than LLMs in both settings, implying that misalignment doesn't stem from the RMs themselves. Additionally, the larger 70B RM displays stronger alignment than the smaller 13B RM, suggesting a positive correlation between RM size and alignment suggests a potential link that motivates further investigation.

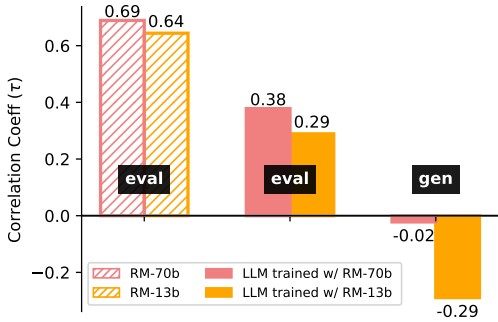

Figure 4: Factor-level preference alignment($\tau$) of human preferences with Reward Model (RM) preferences and LLMs trained with that RM on the summarization task.

**Alignment over Latent Preference.** Our experiments reveal that under single-score human preference, the model can exhibit false positive optimization by producing overly lengthy outputs and misleadingly exhibiting higher alignment scores, similar to Park et al. (2024); Skalse et al. (2022). This is particularly problematic for downstream tasks like summarization, which require concise responses with the original intention well preserved. PROFILE can be used to diagnose latent human preference misalignment and provide training signals to improve alignment at the factor level. Similar to fine-grained RLHF (Wu et al., 2023), we can leverage factor-level scores to align the LLM. Additionally, similar to LLMRefine (Xu et al., 2024), we can employ fine-grained guidance to harness the LLM's self-refinement capability for further improvement.

**Validation of Score-based Generation Approach.** Our research deviates from the typical language model setup by using a 1-5 scoring system for response generation. To assess the validity of our approach, we compare responses generated through direct generation (without scoring) with those across the score range through all summary, helpfulness, and document-based QA tasks. In every

task, we found that score 5 consistently aligns best with direct generation responses, based on the fine-grained factors we use, in models like GPT-4o, Tulu 2.5 + PPO (70B RM), and LLaMA 3.1 70B (see Table 10 in the Appendix H). This suggests that our scoring framework, specifically score 5, captures the essence of unconstrained language model outputs, implying the potential generalizability of our findings to general settings.

**Limitations.** This study has several limitations. First, the preference datasets used may not fully represent the entire spectrum of human preferences. Second, due to budget constraints, human evaluations of model outputs were conducted on a limited scale, with a restricted number of participants, and only on one task. Furthermore, this study represents a preliminary exploration into methods for achieving better alignment, highlighting the potential of various techniques to enhance generation and evaluation. Extensive studies are required to thoroughly assess the efficacy and generalizability of these methods. While this study focuses on post-hoc correction methods, future research should investigate how to incorporate the identified preference factors as signals during the training stage. Additionally, exploring how to embed these signals within datasets used for preference optimization represents a promising direction for future work.

## 6 RELATED WORK

**Explainable Evaluation of LLMs.** Recent research has increasingly emphasized the need for more explainable evaluations of LLMs. For instance, researchers have proposed fine-grained atomic evaluation settings for tasks like fact verification and summarization (Min et al., 2023; Krishna et al., 2023), developed a benchmark for fine-grained holistic evaluation of LLMs on long-form text (Ye et al., 2024), and enhanced evaluation transparency through natural language feedback (Xu et al., 2023). Building on this trend, our work shifts from evaluating individual factors in isolation to analyzing their influence on human preferences and investigating the alignment between human and model judgments regarding the relative importance of these factors. Furthermore, researchers are actively exploring the potential of LLMs as evaluators. Fu et al. (2024); Madaan et al. (2024); Liu et al. (2023) demonstrate the capacity of large models like GPT-4 to achieve human-like system-level evaluation. However, recent works (West et al., 2023; Oh et al., 2024) reveal discrepancies in model performance between generation and evaluation tasks. Inspired by frameworks to meta-evaluate llm as an evaluator (Zheng et al., 2023; Ribeiro et al., 2020), our work evaluates not only the quality of model-generated text but also the alignment of model preferences in evaluation settings, providing a more comprehensive assessment of LLM capabilities.

**Human-AI Preference Alignment.** Aligning large language models (LLMs) with human preferences is a central focus in LLM research, leading to techniques like supervised instruction tuning (Mishra et al., 2021; Wei et al., 2021), RLHF (Ouyang et al., 2022), DPO (Guo et al., 2024), and RLAIF, which utilizes AI-generated feedback (Bai et al., 2022; Lee et al., 2023). However, most studies focus on overall performance (e.g., a response as a whole). While some work has explored using fine-grained human feedback (Dong et al., 2023; Wu et al., 2024), a comprehensive understanding of how granular factors contribute to and differentiate human and model preferences is still lacking. Hu et al. (2023) take a step in addressing this gap by probing the factors influencing human preferences. Building on this work, we expand the investigation of granular preference alignment across multiple tasks and extend the analysis to model generation, providing a comparative analysis of the factors driving both human and model preferences.

## 7 CONCLUSION

We introduce PROFILE, a novel framework for granular factor level analysis of LLM alignment with human preferences. Our analysis using PROFILE reveals that LLMs tend to over-prioritize factors like output length, misaligning human preferences during generation. However, these models exhibit stronger alignment in evaluation tasks, indicating the potential for leveraging evaluative insights to improve generative alignment. By advancing beyond coarse-grained methods, PROFILE facilitates a nuanced understanding of the alignment gaps and mismatches between human and model preferences. These insights underscore the necessity for more sophisticated, factor-level alignment strategies that can guide the development of LLMs to better align with human expectations, ultimately fostering more reliable aligned AI systems.

## 8 ETHICS STATEMENT

Our research relies on established benchmarks and models, and does not involve the development of new data, methodologies, or models that pose significant risks of harm. The scope of our experiments is limited to analyzing existing resources, with a focus on model performance. Human studies conducted within this work adhere to relevant IRB exemptions, and we ensure fair treatment of all participants. Our work is mainly focused on performance evaluation, we recognize that it does not specifically address concerns such as bias or harmful content.

## 9 REPRODUCIBILITY STATEMENT

The datasets and models we use in our study are detailed in § 4.1. For more comprehensive descriptions of the datasets and specific versions of the models, please refer to Appendix C.1 and C.2. The methodology we employed for factor extraction in our experiments is presented in Appendix D, while the prompting configurations set up for the experiments can be found in Appendix E and F. Appendix G and H contain additional experimental results not presented in the main paper. Appendix G provides the lists of all factor scores for both generation and evaluation across all three tasks used in the study. Appendix H presents detailed results regarding the generalizability of our findings in the § 5.

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

APPENDIX

# A   HUMAN EVALUATION OF MODEL GENERATIONS

We collect human preference data via Amazon Mechanical Turk (MTurk) for 30 posts and 6 models. For each post, three summary pairs—selected from five model-generated summaries (scored 1 to 5)—are presented to three annotators. Annotators, restricted to US-based workers with a 95% and HIT approval rate and over 5,000 approved HITs, follow these instructions: *"A good summary is a shorter piece of text that captures the essence of the original. It aims to accomplish the same purpose and convey the same key information as the original post. Please choose the better summary, A or B."* Each annotation is compensated at $0.25. This process yields 1,620 annotations (30 posts * 6 models * 3 pairs/model/post * 3 annotations/pair).

For each model and corresponding summary pair, we calculate an agreement rate, which indicates the percentage of pairs where at least two out of three annotators prefer the summary with the higher score assigned by the model. Each model is evaluated on 90 summary pairs, derived from 30 posts with three pairs per post.

## A.1   EXAMPLES OF HUMAN-MODEL MISALIGNMENT

---

### GPT-4o misaligned examples

**Post:** Good Morning/Afternoon r/advice, Never posted on Reddit before at all, but I figured (based on the overall reliability of you nice individuals) that now would be a good time to start.

About a year ago I moved away from my Midwest town to the west coast for work, and have just recently moved back. I was calling the Utility providers in my area to set up the gas/water/elec for my new place, but I was informed that gas could not be set up due to owing a past amount.

(...)

My question is, am I really on the hook for these fees and what recourse could I possibly have? I cannot afford to pay it, but I also don't think that it should be on me TO pay it. Any advice is truly appreciated as I have not had any experience with this type of thing before.

**GPT-4o's score 5 generation** [length: 93, # facts: 10, src. coverage: 0.389]: A Reddit user recently moved back to their Midwest hometown and, while setting up utilities for their new place, discovered they owe $500 in gas bills from a college house they lived in until 2012. The user didn't cancel the service when they moved out, leading to the bill accruing over time. They are looking for advice on whether they are responsible for this debt and what recourse they might have, as they are unable to afford the payment. The user is new to Reddit and appreciates any guidance on handling this situation.

**GPT-4o's score 3 generation** [length: 61, # facts: 9, src. coverage: 0.44]: A Reddit user seeks advice after discovering they owe $500 in gas bills from a college house they left in 2012. They didn't cancel the service when they moved out, and the bill continued to accrue. They want to know if they are responsible for paying it and what actions they can take, as they can't afford to pay the bill. **(Human Preferred Output)**

---

A higher score summary generated by GPT-4o as shown by the factor-level analysis (in magenta), is longer and includes details not necessary to understand the essence of the original post.

## A.2   HUMAN EVALUATION OF MODEL GENERATIONS

The agreement rates for each model are summarized in Table 4. Despite low factor-level preference alignment ($\tau$), overall agreement rates range from 56% to 75%. This suggests that binary evaluations, where annotators choose the better summary based on overall quality, can mask factor-level misalignments such as preferences for summary length. Since models consistently favor longer responses, continued reliance on this setup to generate human-annotated training data may reinforce the tendency towards verbose answers, aligning with human preferences superficially but missing finer-grained alignment on factors like length or conciseness.

| | $\tau$ | Agree.(%) |
|---|---|---|
| Gemini 1.5 | **0.289** | 0.689 |
| Mixtral | 0.2 | 0.678 |
| GPT-4o | 0.156 | 0.722 |
| LLaMA 3.1 70B | 0.111 | **0.756** |
| Tulu 2.5 + PPO (70B RM) | 0.111 | 0.567 |

Table 4: Factor-level preference alignment ($\tau$) and response-level agreement with human preferences in the summarization Task.

# B ADDITIONAL EXAMPLES OF MODEL GENERATIONS AND EVALUATIONS

## B.1 GENERATION

Table 5 shows summaries of scores 1-5 that GPT-4o generated. We can see a clear trend of higher score summaries being longer, with more information density. The information additional to lower-scoring summaries is the ages of the characters, which is not essential in understanding the original post. This is a randomly selected example and a typical model-generated summary case.

| Score 1 | Score 2 | Score 3 | Score 4 | Score 5 |
|---|---|---|---|---|
| The poster and their fiance have scabies and are treating it. Their roommate refuses to get treated or clean his things, risking reinfestation. The poster is considering asking him to move out. | A couple living with a roommate got scabies and started treatment. The roommate, who has no symptoms, refuses to treat himself or his belongings. The poster is worried about reinfestation and may ask the roommate to move out since he isn't on the lease. | A 19-year-old woman and her 20-year-old fiance contracted scabies after moving into a new apartment with a mutual friend. Both are undergoing treatment, but the roommate refuses to get treated or clean his belongings. The poster is concerned about reinfestation and is contemplating asking the roommate to move out, as he is not on the lease. | A 19-year-old woman and her 20-year-old fiance recently moved into a new apartment with a mutual friend and contracted scabies. Despite both of them undergoing treatment, their roommate refuses to take any preventive measures or treat himself, even though he has not shown symptoms yet. The poster is worried that this will lead to a reinfestation and is considering asking the roommate to move out since he is not on the lease. | A young woman (19) and her fiance (20) recently moved into a new apartment with a mutual friend and unfortunately contracted scabies. Both have started treatment after a doctor's confirmation. However, their 20-year-old roommate, who has not shown symptoms, refuses to undergo treatment or clean his clothes and linen. This refusal has caused concern for the woman, as she fears reinfestation and is seriously considering asking the roommate to move out, especially since he is not on the lease. |

Table 5: An example of GPT-4-generated summaries of scores 1-5. The numbers in the parentheses are the model-assigned scores.

Additionally, we analyzed cases where model-generated outputs received higher scores but are not preferred by human annotators (based on majority votes). The human-model misalignment is more pronounced when comparing summaries with larger score gaps assigned by the models. We provide additional examples of such divergence for GPT-4o, LLaMA 3.1 70B and Tulu 2.5 + PPO (13B RM). We find that when the length and information gap is significant, humans favor summaries that are shorter and less information-dense, as model-generated summaries with lower scores (1-2) generally convey the main intention of the post as well. When factual content is similar, humans tend to favor more informal responses (e.g., "Guy" over "An 18-year-old male", and "gotten back together" over "renewed romantic connection"), which aligns with our factor-level findings indicating higher formality-alignment for human-preferred outputs, since Reddit posts are mostly informal.

---

*Warning: Examples below contain content that may be offensive or upsetting.*

**GPT-4o misaligned examples**

**GPT-4o's score 4 generation**: An 18-year-old male in a long-distance relationship with his 17-year-old girlfriend told her he loves her but wants an open relationship. The girlfriend, who has been with four other boys before him, reacted very negatively, telling him to "FUCK OFF" and that she never wants to speak to him again. He is seeking advice on how to make the situation easier for her and whether he can maintain their relationship while exploring other people.

**GPT-4o's score 1 generation**: Guy told his girlfriend he wants an open relationship. She got mad and told him to go away. He wants advice. **(Human Preferred Output)**

---

**Llama-3.1-70b misaligned examples**

**Llama's score 4 generation**: I'm in a relationship with an 18-year-old girl who's a self-described Tsundere. We share similar interests, but her mood swings are causing problems. She can be affectionate and loving, but also violent and aggressive, lashing out at me over minor issues. For example, she threw shoes at me for a joke and almost stabbed me with a knife over a trivial matter. I'm scared and unsure how to navigate these outbursts.

**Llama's score 2 generation**: I'm dating a girl who's similar to me, but she's really violent and has mood swings. She gets angry over small things and has even attacked me. I'm scared and need help. **(Human Preferred Output)**

---

**Tulu 2.5 + PPO (13B RM) misaligned examples**

**Tulu's score 4 generation**: The OP discovered a potentially incriminating text exchange between their SO and her ex's sister, suggesting a renewed romantic connection.

**Tulu's score 1 generation**: The OP read a suspicious text on their SO's phone, sent from her ex's sister, that suggested they might have gotten back together. **(Human Preferred Output)**

---

## B.2 EVALUATION

We provide examples where the model evaluations align with human preferences, even if the chosen option contains less facts or is shorter. In the first example, where both GPT-4o and LLaMA 3.1 70B correctly chose human-preferred summary, while the chosen summary is shorter, it more accurately reflects the key issue in the original post by mentioning the writer's economic status. In the second example, the GPT-4o chosen summary is more clearly reflecting the content in post over the other option which analogically describes the main idea of the post.

> **GPT-4o & LLaMA aligned examples**
>
> **Post:** Yesterday, I accidentally dropped my Motorola Atrix 2 and the screen cracked really badly. My phone is still fully functional, but it's a bit difficult to see what I'm doing when I'm texting or web browsing, etc. Anyway, I stupidly didn't buy insurance for my phone and I'm not eligible for an upgrade until next May! AT&T offers some options as far as getting a no-commitment phone at a slight discount, but spending $300-$600 for a new phone isn't really in the budget right now.
> (...)
> I found a couple websites that will repair your phone if you send it in. [Doctor Quick Fix] will do it for $110 and I'm still waiting on a quote from [CPR](So my question is, have any of you used this company, or know anyone who has used it? Should I trust these companies? Do you have any recommendations? What should I do to get my phone fixed?
>
> **Summary A**: Dropped my phone, they said they won't repair phones that have been physically abused. Looking for suggestions on cell phone repair companies, if any, and what I should do to get my phone fixed.
>
> **Summary B**: I dropped my phone, cracking the screen. I can't afford to buy a full price phone, so should I try the above repair companies? What should I do? **(Human Preferred Output)**

> **GPT-4o aligned & LLaMA misaligned examples**
>
> **Post:** I got a letter in the mail saying I've been passed up for being hired for my dream job. I wanted this job for 10 damn years and now it's over. I've trained my body, mind, and soul for this job and just through a simple letter, I've been removed from that process. I was in good standing with getting hired. Passed everything with flying colors.
> (...)
> Now what? Am I to live with my parents the rest of my life? Am I to never get my dream car? Am I to just keep my job where I only get paid minimum wage while I make the company tens of thousands? I don't know what to do. I mean my second dream job would be to work with penguins, but I don't think that's possible for me. Anyone have any advice for me? What should I do?
>
> **Summary A**: I followed the yellow brick road for half my life and ended up at a complete dead end and I can't turn around to go back.
>
> **Summary B**: Got passed up for a dream job. Now what the hell are I supposed to do with my life that doesn't include my dream job? **(Human Preferred Output)**

## C EXPERIMENTAL SETTING

### C.1 TASKS

We examine three publicly available datasets of pairwise human judgments commonly used in preference optimization methods like RLHF and DPO training: **Reddit TL;DR** We analyze the dataset released by OpenAI (Stiennon et al., 2020), which includes human ratings of summaries across multiple axes (referred to as "`axis evaluations`"). Higher scores indicate human preference across multiple evaluation dimensions. **StanfordHumanPreference-2 (SHP-2)** (Ethayarajh et al., 2022), focuses on capturing human preferences over responses to questions and instructions, prioritizing helpfulness. Higher scores indicate a more helpful response. For this study, we use responses from the "`reddit/askacademia`" domain. **OpenAI WebGPT** This dataset (Nakano et al., 2021), addresses the task of generating answers to questions from the ELI5 (*"Explain Like I'm Five"*) subreddit. Human annotations compare two model-generated answers based on factual accuracy and overall usefulness. We exclude pairs with Tie ratings in all three datasets, as our analysis focuses on cases with clear preference distinctions.

### C.2 MODELS

Our study focuses on the most advanced and widely-used generative models currently accessible, encompassing both proprietary and open-source options. For open-source models, we include LLaMA

3.1 70B (Dubey et al., 2024)[*], Mixtral 8x7B Instruct v0.1 (Jiang et al., 2024), three TÜLU 2.5 Models (Ivison et al., 2024)—TÜLU 2.5 + PPO 13B (13B RM) [*], TÜLU 2.5 + PPO 13B (70B RM) [*], and TÜLU 2.5 + DPO 13B [*]. For proprietary models, we use Gemini 1.5 Flash (Reid et al., 2024), GPT-4o (OpenAI, 2024) [*], and GPT-3.5 [*]. We set the parameters for all models to: temperature = 0.6, top_p = 0.9, and max_tokens = 1024.

# D PROFILE

## D.1 FACTOR EXTRACTION METHODS

**Rule-based Extraction**  We obtain the Length and Novel Words using a rule-based extraction method. First, we calculate the output's length and count the novel words by removing special characters and splitting the text into words. The total word count represents Length. For Novel Words, we stem both the source text and the model output to create unique sets of stemmed words, then determine the number and proportion of unique words in the output that differ from the source.

**LLM-based Extraction**  The calculations are divided into atomic-fact-level and response-level based on the granularity of the factors.

Atomic-Fact-Level Factors refer to those factors that are evaluated based on the presence or absence of each factor at the atomic fact level. An atomic fact is a short, self-contained piece of information that does not require further explanation and cannot be broken down further (Min et al., 2023). These include the Number Of Facts, Source Coverage, Off Focus, Hallucination, Helpfulness, and Misinformation. The Number Of Facts is determined by counting the total atomic facts, while the remaining factors are calculated as the ratio of relevant atomic facts to the total number of atomic facts.

Response-Level Factors refer to those factors that are evaluated based on the presence or absence of each factor at the response level. These include Receptiveness, Intent Alignment, and Formality Alignment. Formality Alignment is classified into one of three categories: [Aligned/Misaligned/Partially-Aligned], while the other two factors are determined in a binary manner [Yes/No].

The prompts used are provided in D.2. The Source Coverage does not have a separate prompt since it was calculated using the output from the Hallucination (i.e., the ratio of non-hallucinated atomic facts to the total number of atomic facts in the Source Post).

## D.2 PROMPT TEMPLATE FOR LLM-BASED FACTOR EXTRACTION

### D.2.1 TEMPLATE FOR ATOMIC FACT GENERATION

Number Of Fact

> Your task is to extract atomic facts from the INPUT. These are self-contained units of information that are unambiguous and require no further splitting.
>
> {FEW SHOT}
>
> INPUT: input
> OUTPUT:

### D.2.2 TEMPLATE FOR INPUT-OUTPUT FACTORS

Receptiveness

---

[*]Inference for LLaMA was conducted using the Together AI API. https://www.together.ai/

[*]We use huggingface allenai/tulu-v2.5-ppo-13b-uf-mean-13b-uf-rm model.

[*]We use huggingface allenai/tulu-v2.5-ppo-13b-uf-mean-70b-uf-rm model.

[*]We use huggingface allenai/tulu-v2.5-dpo-13b-uf-mean model.

[*]We use gpt-4o-2024-05-13 version for all GPT-4o inference.

[*]We use gpt-3.5-turbo-1106 version for all GPT-3.5 inference.

> Does the response clearly address the query from the original post? First determine the core question or purpose of the original post from the user, and evaluate whether the response clearly serves as the proper answer to the question. Provide your response in JSON format, with a 'yes' or 'no' decision regarding the response's receptiveness to the original post, along with justifications.:
>
> {FEW SHOT}
>
> INPUT:
> Post: {POST}
> Response : {OUTPUT}

### Off Focus

> You have been provided a statement. Can you determine if it is related to the main focus of the post? The main focus of a post is the core subject around which all the content revolves. Format your response in JSON, containing a 'yes' or 'no' decision for each statement in the set, along with justifications.
>
> {FEW SHOT}
>
> INPUT:
> Reddit Post: {POST}

### D.2.3 TEMPLATE FOR SOURCE-OUTPUT FACTORS

### Intent Alignment

> You have been provided a statement. Can you determine if it is related to the main focus of the post? The main focus of a post is the core subject around which all the content revolves. Format your response in JSON, containing a 'yes' or 'no' decision for each statement in the set, along with justifications.
>
> {FEW SHOT}
>
> INPUT: {ATOMIC FACT}
> Reddit Post: {POST}

### Hallucination

> You have been provided with a set of statements. Does the factual information within each statement accurately match the post? A statement is considered accurate if it does not introduce details that are unmentioned in the post, or contradicts the post's existing information. Provide your response in JSON format, with a 'yes' or 'no' decision for each statement in the set, along with justifications.
>
> {FEW SHOT}
>
> INPUT: {ATOMIC FACT}
> Reddit Post: {POST}

### Formality Alignment

> You have been provided an original post and a summary. First determine the formality (formal, informal) for both the post and the summary. Then, decide if the formalities align. If they match perfectly, return "Aligned", if they are similar in terms of formality (e.g., both informal) but have slight differences in how much formal/informal they are, return "Partially Aligned", and if they don't match, return "Not Aligned". Format your response in JSON as follows:
> Output Format: {"decision": , "justification": }
>
> {FEW SHOT}
> Reddit Post: {POST}
> Summary : {OUTPUT}

### D.2.4 TEMPLATE FOR OUTPUT-ONLY FACTORS

### Helpfulness

> You have been provided a statement. Can you determine if this statement provides helpful information, although not directly necessary to answer the question?
>
> {FEW SHOT}
>
> INPUT: question: {POST}
> statements: {ATOMIC FACT}

### Misinformation

> You have been provided a statement. Can you determine if it contains potentially incorrect or misleading information? Potential misleading information include assumptions about user; medical, legal, financial advice; conspiracy theories; claims to take real world action and more.
>
> {FEW SHOT}
>
> INPUT: {ATOMIC FACT}

### D.3 VALIDATION OF LLM-BASED EXTRACTIONS

We use GPT-4o to extract (1) manifestations of response-level factors—Intent Alignment and Formality Alignmentand (2) Number 0f Facts from outputs for our analysis ('atomic-fact-based'). To assess the validity of GPT-4o's evaluation of each factor, we randomly selected 50 samples and found that GPT-4o accurately assessed Intent Alignment in 43 out of 50 samples (86%) and Formality Alignment in 46 out of 50 samples, resulting in an accuracy of 92%. Most misalignments occur when GPT-4o marks a response as 'Not aligned' due to content inaccuracies, even when intent or formality is not the issue. Consistent with prior works using GPT as an extractor of atomic facts (Hu et al., 2023; Min et al., 2023), we find taking atomic facts generated by GPT-4o acceptable and similar to human. We rely on GPT-4o in detecting Hallucination Off Focus, as Hu et al. (2023) reports the accuracy of GPT-4 in these two tasks as 89% and 83%, respectively. Source Coverage is essentially extracted in the same way as Hallucination but with the direction of fact-checking reversed (i.e., checking whether the atomic fact from the source (post) is present in the output (summary)). We further validated GPT-4o's extractions for Helpfulness and Misinformation, finding them largely consistent with human assessments.

For Receptiveness, we randomly sample 50 instances from WebGPT dataset and find the accuracy to be 90%. For Helpfulness, we find the accuracy at a response-level to be 87% and 80% in the atomic-fact-level. The model generally made sound, context-aware judgments, for example, correctly dismissing helpful advice when it contradicted the question's premise (e.g., suggesting coffee when the question stated it didn't help). For Misinformation, we observed 87% response-level accuracy and 70% atomic-fact level precision. Most inaccuracies were false positives, often triggered by exaggerated claims (e.g., "Your paper is now 100% more skimmable").

## E PROMPTS

The details of the model response generation and evaluation prompts we used for each experimental setting are as follows.

### E.1 GENERATION PROMPTS

#### E.1.1 SCORE-BASED GENERATION

The output generation prompts for the three tasks are as follows.

**Task Description**  The following are the descriptions of the three tasks—summarization, helpful response generation, and document-based QA—that are included in the prompt explaining the task to the model. These descriptions replace the {*TASK_DESCRIPTION*} part in each template below.

> - **Summary**: A good summary is a shorter piece of text that captures the essence of the original. It aims to accomplish the same purpose and convey the same key information as the original post.
> - **Heplfulness**: A helpful response is a concise and efficient answer that directly addresses the user's question or task. It should provide accurate and relevant information without unnecessary elaboration.
> - **WebGPT**: A useful answer directly addresses the core question with accurate and relevant information. It should be coherent, free of errors or unsupported claims, and include helpful details while minimizing unnecessary or irrelevant content.

**Generation Template**  The following is the prompt for generating the model's output, rated from 1 to 5, for the given task. The outputs of the three models are referred to as 'summary', 'response', and 'response' respectively. For Tulu and Mixtral models, we customize the prompt by adding ", SCORE 2 SUMMARY:, SCORE 3 SUMMARY:, SCORE 4 SUMMARY:, SCORE 5 SUMMARY:".

```
{TASK_DESCRIPTION} Your job is to generate five [summaries/responses] that would each get a score of 1,2,3,4 and 5.

### Summarization ###
TITLE: {TITLE}
POST: {CONTENT}

### Helpful Response Generation ###
POST: {CONTENT}

### document-based QA ###
Question: {question}
Reference: {reference}

Generate five [summaries/responses] that would each get a score of 1,2,3,4 and 5. SCORE 1 [SUMMARY/RESPONSE]:
```

## E.2  EVALUATION PROMPTS

### E.2.1  COMPARISON-BASED EVALUATION

**Evaluation Template**   We provide the model with two responses using the evaluation prompt below and ask it to assess which output is better. Depending on the task, we also provide relevant sources (e.g., post, question, and reference) along with the responses generated by the model to help it choose the preferred response.

```
{TASK_DESCRIPTION}

### Summarization & Helpful Response Generation ###
Analyze the provided [summaries/responses] and original post, then select the better [summary/response] or indicate if they are equally
good. Output the result in JSON format. Where "better [summary/response]" can be "[Summary/Response] 1", "[Summary/Response]
2", or "Tie" if both [summaries/responses] are equally good.
Output Format:
{{
"better summary": "",
"justification": ""
}}
Reddit Post: {CONTENT}
[Summary/Response] 1: {RESPONSE1}
[Summary/Response] 2: {RESPONSE2}

### document-based QA ###
Where "better answer" can be "Answer 1", "Answer 2", or "Tie" if both responses are equally good.
Question: {QUESTION}

Answer 1: {ANSWER1}
Reference 1: {REFERENCE1}

Answer 2: {ANSWER2}
Reference 2: {REFERENCE2}

Output the result in JSON format.
Output Format:
{{
"better answer": "",
"justification": ""
}}
```

# F  ACHIEVING BETTER ALIGNMENT THROUGH PROFILE

## F.1  IMPROVING ALIGNMENT IN EVALUATION THROUGH FACTOR-LEVEL GUIDANCE.

This section explains the specific experimental settings for the *Improving Alignment in Evaluation through Factor-level Guidance* paragraph in § 4.4. For Guide$_{Mis}$, The Mixtral model we use specified Off Focus as the factor and tulu 2.5 + PPO (13b RM) specified Coherence. These two factors are the ones most preferred by each model but are considered less influential by humans compared to the models. For Guide$_{Rand}$, we randomly select one factor from those that showed no significant preference difference between humans and the models; Fluency is selected for Mixtral, and Off Focus

is selected fortulu 2.5 + PPO (13b RM). The prompts used and the factor-specific guidance included in each prompt are as follows. Prompt template

```
{TASK DESCRIPTION}
{FACTOR SPECIFIC GUIDANCE}
Analyze the provided summaries and original post, then select the better summaries or indicate if they are equally good. Output the
result in JSON format. Where "better summaries" can be "summaries 1", "summaries 2", or "Tie" if both summaries are equally good.
Output Format:
{
"better summary": "",
"justification": ""
}
Reddit Post: {CONTENT}
Summary 1: {RESPONSE1}
Summary 2: {RESPONSE2}
```

Factor Specific Guidance

```
Off Focus: Note that the summary should capture the main focus of the post, which is the core subject around which all the content
revolves.
Hallucination: Note that the summary should contain factual information that accurately matches the post.
Coherence: Note that whether all the sentences form a coherent body or not is not the primary factor in determining the quality of a
summary.
Fluent: Note that the summary should be fluent.
Intent Alignment: Focus on how well the summary represents the main intents of the original post.
```

## F.2 Leveraging Evaluation for Better Alignment in Generation.

### F.2.1 Prompts for Improvement

The prompts we used to enhance the model's output are as follows. We focuses on the Summary task for the experiment.

**Task Description** For Summary task, the description is the same as the one used in the score-based generation prompt.

```
Summary: A good summary is a shorter piece of text that captures the essence of the original.
```

The three prompts used for improvement are as follows.

**Improvement Template**

```
{TASK_DESCRIPTION} It aims to accomplish the same purpose and convey the same key information as the original post. Based on
the evaluation results, improve the summary by addressing the feedback provided.
Reddit Post: {CONTENT}
Summary 1: {SUMMARY1}
Summary 2: {SUMMARY2}
Evaluation: {EVALUATION}
ImprovedSummary/Response:
```

**Improvement Baseline Template**

```
{TASK_DESCRIPTION} Improve the given summary.
Reddit Post: {CONTENT}
Summary: {SUMMARY}
Improved Summary:
```

**Improvement Baseline Single Template**

```
{TASK_DESCRIPTION} Generate an improved summary based on the given two summaries.
Reddit Post: {CONTENT}
Summary 1: {SUMMARY1}
Summary 2: {SUMMARY2}
Improved Summary:
```

### F.2.2 Metric

Due to the relative nature of preference, we cannot directly assess the alignment of the improved response itself. Instead, we measure the degree of the *improvement* resulting from the evaluator's

feedback to evaluate how well the occurred improvement aligns with both human and evaluator preferences. For each factor $f_k$ and pairwise factor comparison function $M_k$, we calculate the *factor score of improvement* with $\tau_{14}$.

For a given initial response $r_{init}$ and the improved response $r_{post}$, since the model is considered to have 'improved' the responses, $r_{post}$ is regarded as the model's 'preferred' response over $r_{init}$. The factor scores are then calculated as follows:

$$\tau_{14}(f_k) = \frac{|C_k| - |D_k|}{|C_k| + |D_k| + |T_k|} \tag{1}$$

where

$$C_k = \sum_{r_{init}, r_{post} \in R} \mathbb{1}[M_k(r_{post}, r_{init}) = +1],$$

$$D_k = \sum_{r_{init}, r_{post} \in R} \mathbb{1}[M_k(r_{post}, r_{init}) = -1],$$

$$T_k = \sum_{r_{init}, r_{post} \in R} \mathbb{1}[M_k(r_{post}, r_{init}) = 0],$$

For the Length factor, if the model produces responses that are longer than the original responses $r_{init}$, (i.e. $M_{\text{length}}(r_{post}, r_{init}) = 1$), this response pair is classified as concordant and vice versa. When evaluating all response pairs, a positive factor score suggests that the model significantly considers this factor when improving responses, while a negative score indicates a negative influence. A score near zero implies that the factor has minimal impact on the improvement process. The magnitude of the score reflects the degree of influence this factor exerts on the response enhancement.

Subsequently, we calculate Kendall's $\tau$ between the set of "factor scores of improvement" for each factor and the factor scores assigned by both human evaluators and automated evaluators, which we denote as $\Delta\tau$. This $\Delta\tau$ quantifies how the model's improvements correlate with human and evaluator's factor-level preferences.

### F.2.3 FEEDBACK VALIDATION

One of the authors examine 30 samples of GPT-4o evaluator's feedback to determine whether it correspond to our predefined factors. The analysis reveals that out of the 30 samples, the most frequently addressed factor in GPT-4o's feedback is Intent Alignment, appearing 20 times. This is followed by Source Coverage, which appeared 15 times, and Number of Facts with 12 occurrences. The Length and Off Focus factors are mentioned 10 and 9 times each. Less frequently addressed is Coherence, which appeared 6 times, and Fluency, which is mentioned 3 times. Factors other than these are not mentioned in the feedback at all. As shown in Table 3 (a), in the evaluation setting, GPT-4o exhibit correlations close to zero or negative for most factors except for Intent Alignment, Formality Alignment, Number of Facts Source Coverage, Length and Coherence. This observed trend aligns with our findings from the feedback, with the exception of Formality Alignment.

## G FACTOR-LEVEL PREFERENCE ALIGNMENT

### G.1 FACTOR SCORES

Table 6- 8 present the full lists of factor scores for both generation (gen) and evaluation (eval) across all three tasks used in the study.

### G.2 FACTOR-LEVEL ALIGNMENT WITH HUMAN AND MODELS.

Table 9 shows models' factor-level alignment (Kendall's $\tau$ ) with humans for helpful response generation tasks (SHP-2) and document-based QA tasks (WebGPT), and response-level agreement with humans in an evaluation setting.

| Factors | Gemini 1.5 | | GPT-3.5 | | GPT-4o | | LLaMA 3.1 70B | | Human |
|---|---|---|---|---|---|---|---|---|---|
| | gen | eval | gen | eval | gen | eval | gen | eval | - |
| intent-align. | 0.208 | **0.681** | 0.092 | **0.463** | 0.142 | 0.626 | 0.227 | 0.650 | **0.596** |
| formality-align. | 0.114 | 0.677 | 0.086 | 0.428 | 0.169 | **0.770** | 0.186 | **0.722** | 0.594 |
| # facts | 0.708 | 0.367 | 0.268 | 0.223 | 0.844 | 0.362 | 0.862 | 0.279 | 0.328 |
| src-cov | 0.640 | 0.384 | 0.234 | 0.224 | 0.779 | 0.339 | 0.880 | 0.361 | 0.274 |
| length | **0.904** | 0.450 | **0.472** | 0.280 | **0.976** | 0.386 | **0.995** | 0.378 | 0.257 |
| coherence | 0.114 | 0.257 | -0.004 | 0.222 | 0.492 | 0.258 | 0.586 | 0.249 | 0.180 |
| off-focus | -0.015 | 0.014 | 0.013 | -0.029 | -0.034 | -0.005 | -0.019 | 0.051 | 0.050 |
| hallucination | 0.075 | -0.120 | -0.001 | -0.054 | 0.058 | -0.106 | 0.004 | -0.130 | -0.037 |
| fluency | -0.165 | -0.011 | -0.081 | 0.012 | -0.012 | -0.033 | 0.227 | -0.087 | -0.072 |
| novel words | 0.534 | -0.088 | 0.318 | -0.107 | 0.508 | -0.213 | 0.354 | -0.091 | -0.167 |

(a) Results Of Gemini 1.5, GPT-3.5, GPT-4o, and LLaMA 3.1 70B

| Factors | Mixtral | | Tulu 70B RM | | Tulu 13B RM | | Tulu DPO | | Human |
|---|---|---|---|---|---|---|---|---|---|
| | gen | eval | gen | eval | gen | eval | gen | eval | - |
| intent-align. | 0.118 | **0.120** | 0.104 | **0.193** | 0.045 | **0.102** | 0.087 | **0.152** | **0.596** |
| formality-align. | 0.086 | 0.038 | 0.018 | 0.183 | -0.002 | 0.081 | 0.102 | 0.120 | 0.594 |
| # facts | 0.588 | 0.073 | 0.409 | 0.075 | 0.322 | 0.039 | 0.383 | 0.078 | 0.328 |
| src-cov | 0.445 | 0.055 | 0.294 | 0.136 | 0.191 | 0.069 | 0.317 | 0.105 | 0.274 |
| length | **0.785** | 0.044 | **0.620** | 0.109 | **0.512** | 0.048 | **0.528** | 0.092 | 0.257 |
| coherence | 0.105 | 0.106 | 0.057 | 0.162 | -0.047 | 0.114 | -0.029 | 0.121 | 0.180 |
| off-focus | 0.028 | 0.144 | 0.003 | -0.046 | -0.011 | -0.053 | 0.011 | -0.044 | 0.050 |
| hallucination | 0.108 | -0.053 | 0.066 | -0.109 | 0.084 | -0.076 | 0.027 | -0.104 | -0.037 |
| fluency | 0.021 | 0.051 | 0.011 | 0.025 | 0.092 | 0.016 | -0.002 | -0.004 | -0.072 |
| novel words | 0.407 | -0.041 | 0.391 | -0.052 | 0.390 | -0.029 | 0.329 | -0.039 | -0.167 |

(b) Results Of Mixtral and Tulu 2.5 Models

Table 6: Full lists of factor scores in generation (gen) and evaluation (eval) in Summarization task. Sorted based on the human factor score.

## G.3 FACTOR CORRELATIONS

Figure 5 presents the correlation matrix for the GPT-4o, Gemini-1.5, and Tulu 2.5 + PPO (13B RM) models across three tasks. The analysis focuses on the correlation between the distributions of feature scores for each feature within the samples generated by these models.

In summarization task, the patterns of feature correlation are generally consistent across the three models. Notably, there is a strong correlation between {length and number of facts} as well as {number of facts and source coverage}. These results are intuitive: the more factual content an answer includes, the longer the response tends to be, which in turn increases the likelihood of covering information from the source material.

In helpfulness task, All three models consistently exhibit a high correlation among {length, number of facts, and helpfulness}. This is expected, as longer responses are more likely to include a greater number of facts, which often translates into more helpful content. Interestingly, in the GPT-4o model specifically, there is a noticeable correlation between "receptiveness" and the set of factors {helpfulness, number of facts, coherence, length}. As detailed in Table 7, these are precisely the factors that GPT-4o tends to prioritize in this task. This pattern suggests that the GPT-4o model frequently considers these factors during response generation, resulting in a higher prevalence of these features in its outputs.

In the WebGPT task, there was a high correlation among {length, number of facts, and helpfulness}, similar to the helpfulness task. For GPT-4o and Tulu 2.5 + PPO (13B RM), the correlation between

| Factors | Gemini 1.5 | | GPT-3.5 | | GPT-4o | | LLaMA 3.1 70B | | Human |
|---|---|---|---|---|---|---|---|---|---|
| | gen | eval | gen | eval | gen | eval | gen | eval | |
| receptive | 0.499 | 0.152 | 0.098 | 0.360 | 0.552 | 0.190 | 0.551 | 0.151 | 0.248 |
| helpfulness | 0.736 | 0.071 | 0.375 | 0.199 | 0.899 | 0.095 | 0.835 | 0.064 | 0.193 |
| # facts | 0.569 | 0.062 | 0.371 | 0.148 | 0.857 | 0.081 | 0.751 | 0.054 | 0.162 |
| length | 0.918 | 0.058 | 0.643 | 0.143 | 0.964 | 0.072 | 0.997 | 0.048 | 0.151 |
| coherent | 0.507 | 0.057 | 0.134 | 0.164 | 0.732 | 0.068 | 0.582 | 0.048 | 0.113 |
| misinformation | 0.061 | 0.036 | -0.012 | 0.039 | -0.131 | 0.036 | 0.150 | 0.031 | 0.089 |
| fluency | -0.088 | 0.058 | 0.112 | 0.078 | 0.095 | 0.060 | 0.077 | 0.056 | 0.088 |
| off-focus | 0.013 | 0.021 | 0.024 | 0.029 | 0.034 | 0.033 | -0.019 | 0.025 | 0.002 |
| hallucination | 0.092 | -0.042 | 0.075 | -0.107 | -0.212 | -0.060 | 0.235 | -0.033 | -0.074 |

(a) Results Of Gemini 1.5, GPT-3.5, GPT-4o, and LLaMA 3.1 70B

| Factors | Mixtral | | Tulu 70B RM | | Tulu 13B RM | | Tulu DPO | | Human |
|---|---|---|---|---|---|---|---|---|---|
| | gen | eval | gen | eval | gen | eval | gen | eval | |
| receptive | 0.413 | 0.133 | 0.059 | 0.132 | 0.063 | 0.132 | 0.163 | 0.105 | 0.248 |
| helpfulness | 0.817 | 0.047 | 0.561 | 0.045 | 0.561 | 0.045 | 0.222 | 0.061 | 0.193 |
| # facts | 0.805 | 0.034 | 0.577 | 0.032 | 0.076 | 0.033 | 0.687 | 0.073 | 0.162 |
| length | 0.946 | 0.033 | 0.822 | 0.031 | 0.822 | 0.030 | 0.862 | 0.062 | 0.151 |
| coherent | 0.561 | 0.039 | 0.171 | 0.037 | 0.161 | 0.036 | 0.295 | 0.061 | 0.113 |
| misinformation | 0.022 | 0.028 | -0.026 | 0.023 | -0.024 | 0.025 | 0.016 | 0.050 | 0.089 |
| fluency | -0.009 | 0.046 | 0.061 | 0.044 | 0.092 | 0.043 | 0.237 | 0.016 | 0.088 |
| off-focus | -0.012 | 0.034 | 0.008 | 0.029 | 0.007 | 0.033 | 0.013 | 0.043 | 0.002 |
| hallucination | -0.021 | -0.027 | 0.110 | -0.027 | 0.202 | -0.026 | 0.132 | -0.060 | -0.074 |

(b) Results Of Mixtral and Tulu 2.5 Models

Table 7: Full lists of factor scores in generation (gen) and evaluation (eval) in SHP2 dataset. Sorted based on the human factor score.

novel word and hallucination was high, which can be explained by the tendency to use novel words when hallucinating something.

## H    GENERALIZABILITY OF OUR RESULTS

We conduct experiments by prompting the model to generate responses with scores ranging from 1 to 5. This setup allows us to verify whether the results can generalize to a typical scenario where the model generates responses directly. We compare the model's direct responses and the score-based responses for the summarization task on Reddit TL;DR using outputs from GPT-4o, Tulu 2.5 + PPO (70B RM), and LLaMA 3.1 70B.

Since the value ranges differ across features, we scale the data using min-max scaling before calculating cosine similarity. The results in Table 10 indicate that the model's direct responses are most similar to those with a score of 5, all showing a high similarity of over 0.85. Overall, as the scores decrease, the similarity also declines.

This finding suggests that the model's direct responses align closely with its best-generated responses. Additionally, the lower the score, the less similarity there is to the direct responses, indicating that our score-based responses align well with the model's outputs. Thus, we demonstrate that our findings can generalize to typical settings where responses are generated directly by the model.

| Factors | Gemini 1.5 gen | eval | GPT-3.5 gen | eval | GPT-4o gen | eval | LLaMA 3.1 70B gen | eval | Human |
|---|---|---|---|---|---|---|---|---|---|
| receptive | 0.422 | 0.255 | 0.119 | 0.144 | 0.407 | 0.324 | 0.493 | 0.209 | 0.362 |
| length | 0.965 | 0.129 | 0.660 | 0.033 | 0.965 | 0.048 | 0.981 | 0.111 | 0.092 |
| helpfulness | 0.328 | 0.120 | 0.157 | 0.027 | 0.182 | 0.046 | 0.178 | 0.056 | 0.085 |
| # facts | 0.304 | 0.128 | 0.258 | 0.001 | 0.091 | 0.056 | -0.026 | 0.047 | 0.072 |
| coherence | 0.780 | 0.069 | 0.483 | 0.030 | 0.865 | 0.047 | 0.771 | 0.056 | 0.067 |
| fluency | 0.140 | -0.001 | 0.017 | 0.044 | 0.170 | 0.045 | 0.302 | 0.016 | 0.043 |
| misinformation | 0.146 | -0.059 | 0.005 | -0.005 | -0.073 | -0.089 | 0.110 | -0.003 | -0.002 |
| off-focus | 0.018 | 0.018 | 0.002 | 0.036 | 0.027 | 0.036 | 0.017 | 0.082 | -0.023 |
| novel_words | 0.211 | -0.056 | 0.205 | 0.012 | 0.093 | -0.031 | -0.346 | -0.016 | -0.053 |
| hallucination | 0.025 | -0.083 | -0.013 | 0.000 | -0.200 | -0.098 | -0.229 | -0.045 | -0.139 |

(a) Results Of Gemini 1.5, GPT-3.5, GPT-4o, and LLaMA 3.1 70B

| Factors | Mixtral-eval gen | eval | Tulu 70B RM gen | eval | Tulu 13B RM gen | eval | Tulu DPO gen | eval | Human |
|---|---|---|---|---|---|---|---|---|---|
| receptive | 0.313 | 0.064 | 0.086 | 0.129 | 0.093 | 0.144 | 0.183 | 0.202 | 0.362 |
| length | 0.874 | -0.019 | 0.033 | 0.884 | 0.014 | 0.844 | 0.101 | 0.856 | 0.092 |
| helpfulness | 0.276 | 0.002 | 0.021 | -0.041 | 0.028 | 0.047 | 0.083 | 0.558 | 0.085 |
| # facts | 0.251 | -0.042 | -0.015 | -0.042 | -0.010 | 0.067 | 0.065 | 0.057 | 0.072 |
| coherence | 0.776 | 0.010 | -0.007 | 0.504 | 0.003 | 0.491 | 0.018 | 0.617 | 0.067 |
| fluency | 0.048 | 0.026 | 0.030 | 0.105 | 0.038 | 0.133 | 0.006 | 0.054 | 0.043 |
| misinformation | 0.157 | 0.018 | 0.017 | 0.131 | -0.012 | 0.050 | 0.018 | 0.157 | -0.002 |
| off-focus | 0.038 | 0.024 | 0.025 | -0.021 | 0.013 | 0.016 | 0.028 | 0.015 | -0.023 |
| novel_words | -0.094 | 0.004 | 0.026 | 0.422 | 0.010 | 0.396 | 0.003 | 0.193 | -0.053 |
| hallucination | -0.130 | 0.025 | 0.018 | 0.096 | 0.003 | 0.043 | -0.023 | -0.017 | -0.139 |

(b) Results Of Mixtral and Tulu 2.5 Models

Table 8: Full lists of factor scores in generation (gen) and evaluation (eval) on document-based QA tasks (WebGPT). Sorted based on the human factor score.

| | Generation $\tau$ | Evaluation $\tau$ | Agree.(%) | Generation $\tau$ | Evaluation $\tau$ | Agree.(%) |
|---|---|---|---|---|---|---|
| GPT-4o | 0.556 | **0.944** | 0.819 | 0.60 | 0.778 | **0.654** |
| Gemini 1.5 | 0.444 | 0.889 | 0.846 | 0.60 | **0.822** | 0.61 |
| GPT-3.5 | 0.389 | 0.833 | 0.721 | 0.467 | 0.378 | 0.551 |
| LLaMA 3.1 70B | 0.5 | 0.722 | **0.845** | 0.60 | 0.689 | 0.605 |
| Tulu 2.5 + PPO (70B RM) | 0.222 | 0.611 | **0.845** | 0.067 | 0.200 | 0.520 |
| Tulu 2.5 + PPO (13B RM) | 0.056 | 0.556 | 0.844 | 0.333 | 0.378 | 0.526 |
| Mixtral | **0.667** | 0.556 | 0.845 | **0.778** | -0.200 | 0.529 |
| Tulu 2.5 + DPO (13B) | 0.511 | 0.809 | 0.684 | 0.333 | 0.667 | 0.540 |

(a) Helfulness      (b) document-based QA

Table 9: Model correlations (Kendall's $\tau$) with human values for helpful response generation tasks (SHP-2) and document-based QA tasks (WebGPT), and response-level agreement with human preferences.

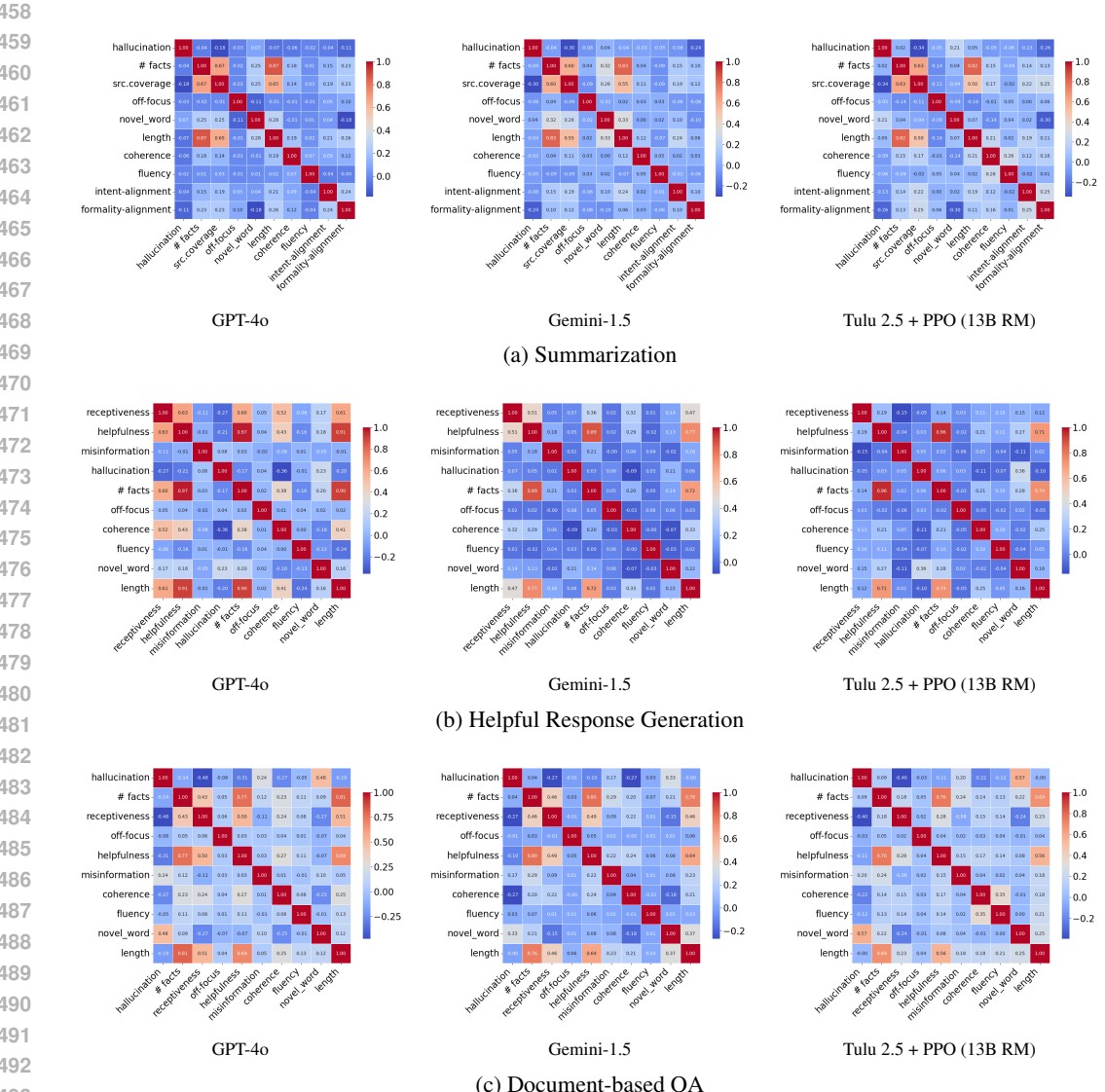

Figure 5: Correlation matrices for various models across tasks.

| Task | Model | Score 1 | Score 2 | Score 3 | Score 4 | Score 5 |
|---|---|---|---|---|---|---|
| Summarization | GPT-4o | 0.791 | 0.823 | 0.856 | 0.886 | **0.901** |
| | Tulu 2.5 + PPO (70B RM) | 0.831 | 0.852 | 0.850 | 0.856 | **0.863** |
| | LLaMA 3.1 70B | 0.711 | 0.792 | 0.828 | 0.849 | **0.854** |
| Helpful Response Generation | GPT-4o | 0.532 | 0.604 | 0.620 | 0.637 | **0.685** |
| | Tulu 2.5 + PPO (70B RM) | 0.435 | 0.492 | 0.581 | 0.641 | **0.679** |
| | LLaMA 3.1 70B | 0.463 | 0.516 | 0.628 | 0.662 | **0.690** |
| Document-based QA | GPT-4o | 0.528 | 0.599 | 0.625 | 0.657 | **0.697** |
| | Tulu 2.5 + PPO (70B RM) | 0.513 | 0.572 | 0.631 | 0.691 | **0.738** |
| | LLaMA 3.1 70B | 0.532 | 0.570 | 0.644 | 0.706 | **0.765** |

Table 10: Comparison of similarity between directly generated responses and score-based responses for summarization, helpful response generation, and document-based QA tasks.

