# OpenReview forum: "Discovering Factor Level Preferences to Improve Human-Model Alignment"
_ICLR.cc/2025/Conference — Submitted to ICLR 2025_

### Official Review · Reviewer_vTJd · 2024-11-04

**Soundness:** 3
**Presentation:** 3
**Contribution:** 3
**Rating:** 6
**Confidence:** 4

**Summary:**

This paper introduces PROFILE, a framework designed to uncover and quantify the specific factors influencing both human and LLM preferences in language generation tasks. It addresses the problem of misalignment between LLM outputs and human preferences by providing a granular, factor-level analysis rather than relying on coarse-grained comparisons. The main contributions include the development of PROFILE, its application across three tasks (TLDR summarization, helpful response generation, and WebGPT document-based QA), and demonstrating how factor-level insights can improve human-LLM alignment

**Strengths:**

● The paper presents an explainable framework that enhances understanding of human-model preference alignment at a granular level.
● It addresses a significant gap in current methods by focusing on specific factors influencing preferences, which can guide improvements in LLM training.
● Demonstrating that leveraging factor-level insights can improve alignment has practical implications for developing more human-aligned LLMs.

**Weaknesses:**

● The paper might not thoroughly compare with existing methods, leaving questions about its relative advantages.
● The paper may lack sufficient empirical validation due to limited experiments or datasets, potentially affecting the generalizability of its conclusions.
● There might be concerns about the scalability of the proposed framework without fine-grained human annotations, impacting its practicality.

**Questions:**

1. How does your approach compare quantitatively and qualitatively with existing methods in preference alignment? Such as all kinds of llm-as-a-judge methods / G-Eval / ... etc

2. Can you provide more details on how the framework performs when applied to tasks beyond the three studied, and are there limitations to its generalizability? Such as creative writing, role-playing, and coding.

3.  The boundary between Receptiveness / Intent Align / Helpfulness is vague and not independent of each other.

---

> ### Author Response · Authors · 2024-11-22
>
> We thank the reviewer for recognizing the strengths of our work, including the presentation of **"an explainable framework that enhances understanding of human-model preference alignment"** and addressing **"a significant gap in current methods by focusing on specific factors influencing preferences."** We appreciate the acknowledgment that our approach has **"practical implications for developing more human-aligned LLMs."** In response to the reviewer's valuable feedback, we have incorporated revisions to further clarify and enhance the manuscript, marked in red. Below, we address specific comments and suggestions in detail.
>
> ---
>
> ###  **“might not thoroughly compare with existing methods, leaving questions about its relative advantages”, “... compare quantitatively and qualitatively with existing methods in preference alignment? Such as all kinds of llm-as-a-judge methods / G-Eval / ... etc”**
>
> **_(CLARIFICATION)_** Existing methods like LLM-as-a-judge and G-Eval primarily focus on evaluating how well a model's outputs meet predefined criteria, aiming to score or rank responses based on absolute quality metrics such as fluency, coherence, and relevance.
>
> PROFILE diverges fundamentally from these approaches because its goal is to interpret what the model values when generating or evaluating better outputs. Rather than simply assessing output quality, we investigate why certain outputs are deemed better by uncovering the underlying preference mechanisms.
> - **Qualitative Difference**: PROFILE seeks to understand preference alignment by uncovering the importance of different criteria (e.g., fluency, coherence) in the model's own judgments between high-quality and low-quality outputs. For example:
>     - LLM-as-a-judge evaluates how "fluent" an output is.
>     - PROFILE evaluates how fluency is differentially represented in outputs the model prefers versus those it does not, revealing the model's internal prioritization.
> - **Quantitative Advantage**: PROFILE's methodology enables the analysis of relative differences between the factors, highlighting which criteria significantly influence the model's preferences. This offers a novel lens for interpretability not captured by traditional evaluation methods.
>
> ---
>
> ### **"lack sufficient empirical validation due to limited experiments or datasets"**
>
> **_(CLARIFICATION)_** While we appreciate the reviewer's concern regarding the limited dataset coverage, we want to clarify that we intentionally chose three representative tasks—summarization, helpfulness, and document QA—because the factors in these tasks encompass core aspects of preference alignment tasks, making our findings broadly relevant to similar tasks.
>
> ---
>
> ### **"Can you provide more details on how the framework performs when applied to tasks beyond the three studied, and are there limitations to its generalizability? Such as creative writing, role-playing, and coding."**
>
> **_(CLARIFICATION)_** We deliberately excluded tasks such as coding and mathematics from our evaluation because these domains are primarily assessed by output "correctness" rather than nuanced preference alignment, which falls outside the core objective of our study.
>
> However, PROFILE's methodology is inherently extensible to other domains such as creative writing and role-playing. The process is straightforward:
>
> 1. Define additional factors (e.g., "plot originality" for creative writing).
> 2. Establish definitions and prompts for evaluating these new factors.
> 3. Extract factor-level preferences and analyze them using our metrics as outlined in the paper.
>
> This demonstrates PROFILE's potential for broad applicability beyond our initial tasks.
>
> ---
>
> ### **"concerns about the scalability of the proposed framework without fine-grained human annotations, impacting its practicality."**
>
> Thank you for your thoughtful review and careful consideration of our work. We appreciate the opportunity to provide clarification.
>
> **_(CLARIFICATION)_**  Far from being a limitation, the absence of fine-grained human annotations is actually a key advantage of our proposed framework. As explained in **lines 122-123** ("Specifically, we experiment with a pairwise preference setting, where preferences are determined by comparing two responses"), our methodology requires only coarse-grained preference information from both human and model responses. As detailed in **Section 3.2**, PROFILE compares response-level preferences with factor manifestations extracted automatically to uncover each factor's influence. In other words, automatically analyzing the impact of fine-grained factors is the core of our PROFILE methodology.
>
> **_(REVISION)_** To address potential misunderstandings, we have enhanced the clarity and organization of **Sections 2 and 3** in the draft, providing more precise explanations of our approach.
> We hope the reviewers will reconsider their concerns in light of these clarifications and revisions.

---

> ### Author Response · Authors · 2024-11-23
>
> ### **“The boundary between Receptiveness / Intent Align / Helpfulness is vague and not independent of each other.”**
>
> **(CLARIFICATION)** The distinctions between Intent-Alignment, Receptiveness, and Helpfulness are more nuanced than they might initially appear:
> Intent-Alignment and Receptiveness differ in their comparative approach: one compares to the source, while the other examines input-output relationships. Because of this inherent difference in the tasks they are considered, they are not measured at the same time, so they are independent of each other.
> Receptiveness and Helpfulness are defined distinctly as Input-Output vs. Output-only metrics, and have been treated differently in prior research. Specifically, while receptiveness measures “Whether the core question of the input has been answered,” Helpfulness measures “The ratio of facts that provide additional helpful information.”
>
> **(ADDITIONAL ANALYSIS)** Our additional feature correlation analysis (Appendix G.3, Figure 5) supports these distinctions. We analyzed the correlation of feature score distributions across model-generated samples
> Receptiveness and Helpfulness showed low correlation across models:
> - Highest correlation: GPT-4o at 0.63
> - Lowest correlation: Tulu-PPO-13B at 0.19
>
> Since the correlation between the factors is not consistently high and varies depending on the model, these results suggest that even if the factors in GPT-4o exhibit high correlation, it is not due to the inherent characteristics of the factors themselves, but rather to the model's behavior.

---

> ### Author Response · Authors · 2024-11-25
> **Have you read our comments?**
>
> Dear Reviewer  vTJd,
>
> We understand the discussion timeline is inconvenient for your busy schedule, but we would love to continue our discussion before the time window closes.
>
> We hope that we were able to resolve all your questions and please let us know if there's more.
>
> Best,\
> authors

---

> ### Comment · Reviewer_vTJd · 2024-11-26
> **comments**
>
> Thank you for your detailed response. In the absence of evidence for further generalizability, I will keep my score the same.

---

### Official Review · Reviewer_ZMM8 · 2024-11-04

**Soundness:** 2
**Presentation:** 2
**Contribution:** 3
**Rating:** 5
**Confidence:** 4

**Summary:**

This paper introduces a framework called PROFILE, designed for fine-grained factor analysis of LLM alignment. PROFILE reveals the discrepancies between human and AI preferences, pinpointing specific areas of divergence. By quantifying the influence of various factors, this framework provides new insights into the interpretability and improvement of models.

**Strengths:**

- **Comprehensive Analysis from a Unique Perspective**: The paper conducts a detailed analysis of the underlying factors affecting human and AI preferences, providing a comprehensive view into the mechanisms of preference alignment.
- **Interpretability**: By performing factor-level preference analysis, PROFILE helps identify specific reasons for human-model preference divergence, offering clear directions for model optimization.

**Weaknesses:**

- **Unclear Justification for Multi-level Factor Classification**: While the paper proposes a three-level factor classification system, it does not sufficiently explain the basis for each factor level, the scientific soundness of the classification, or its handling of task complexity. This raises concerns about whether the framework can accurately reflect human-model preference differences.
- **Potential Issues in the Analytical Approach**: The study uses GPT-4o for factor quantification while examining human-AI preference alignment, which may introduce new biases, potentially affecting the objectivity of the analysis.
- **Limitations in Experimental Design**: The paper validates the PROFILE framework on a limited set of public datasets, which restricts its demonstration of applicability to other tasks. Moreover, it lacks sufficient ablation studies to analyze the contribution of each factor, making it difficult to understand their impact across tasks.
- **Weak Correspondence between Results and Conclusions**: Although the experiments showcase preference alignment in some tasks, they lack clear methodological and empirical support for guiding improvements in human-AI alignment. The paper does not clarify how PROFILE contributes to enhancing model performance or its impact on generation quality in practical applications.
- **Over-reliance on Quantitative Metrics in Analysis**: The paper's analysis mostly depends on correlation and quantitative scores, lacking qualitative insights into why the model exhibits inconsistencies with human preferences for certain factors. This approach results in a somewhat superficial view that fails to reveal the deeper reasons behind the observed divergences.

**Questions:**

1. I am curious about the correlations between different factors. Could you provide an analysis on this?

---

> ### Author Response · Authors · 2024-11-19
>
> We thank reviewer ZMM8 for their feedback, for recognizing the core value of our work in **"identifying specific reasons for human-model preference divergence,"** and for finding our analysis **“comprehensive.”** We would like to note that key changes in the updated draft are highlighted in red. Below, we address the reviewer’s specific concerns:
>
> ---
> ### **"uses GPT-4o for factor quantification while examining human-AI preference alignment, which may introduce new biases"**
>
> **_(CLARIFICATION)_** We appreciate the reviewer's concern about potential biases introduced by using GPT-4 for factor quantification. Our decision to use model-based extraction was primarily driven by the need for scalability, while building upon established precedent from [1], which demonstrated the reliability of GPT-4 for similar factor extraction tasks.
>
> While we acknowledge the inherent limitations of model-based evaluation, we took several steps to validate and ensure reliability. Detailed documentation of our validation process can be found in **Appendix D.3**. Specifically,
> 1. We drew on existing research confirming GPT’s capabilities in atomic fact extraction [1] and factor judgment [2].
> 2. We validated our prompts through iterative refinement and human verification, with strict reliability thresholds (>80% alignment with human judgment) before incorporating model-based judgments.
>
> We hope the reviewers will reconsider their concerns in light of these clarifications and our comprehensive validation efforts.
>
> ---
> ### **"validates the PROFILE framework on a limited set of public datasets, which restricts its demonstration of applicability to other tasks"**
>
> **_(CLARIFICATION)_** While we appreciate the reviewer's concern regarding the limited dataset coverage, we want to clarify that PROFILE is designed as a generalizable **methodology** rather than a task-specific framework. We intentionally chose three representative tasks—summarization, helpfulness, and document QA—because the factors in these tasks encompass core aspects of preference alignment tasks, making our findings broadly relevant to similar tasks.
>
> Furthermore, PROFILE's methodology is inherently extensible to other domains. The process is straightforward:
> 1. Define additional factors (e.g., "plot originality" for creative writing).
> 2. Establish definitions and prompts for evaluating these new factors.
> 3. Extract factor-level preferences and analyze them using our metrics as outlined in the paper.
>
> This demonstrates PROFILE's potential for broad applicability beyond our initial tasks.
>
> ---
>
> ### **“lacks sufficient ablation studies to analyze the contribution of each factor”… “impact across tasks”**
>
> **_(CLARIFICATION)_** We want to emphasize that PROFILE is inherently designed to analyze the individual contribution of each factor, making this factor-level analysis a core feature of our framework.
>
> Our results in **Sections 4.1 and 4.2** demonstrate how each factor impacts preference alignment across two scenarios: (1) when models act as generators and (2) when models serve as evaluators.
> The impact of individual factors is illustrated in **Figure 3 (a), (b), and (c)**, highlighting their contributions across different tasks. For deeper insights, we provide extensive factor-level breakdowns in **Appendix G.1, including Table 5 (Summarization), Table 6 (SHP-2), and Table 7 (WebGPT)**.
>
> We believe this detailed factor analysis serves the same purpose as traditional ablation studies by revealing the contribution of each component. Nevertheless, we welcome specific suggestions for additional ablation studies that could further strengthen our analysis.
>
> ---
> ### **“The paper does not clarify how PROFILE enhances model performance or impacts generation quality in practical applications”**
>
> **_(CLARIFICATION)_** We appreciate the reviewer's concern regarding the practical impact of PROFILE on model performance and generation quality. We believe that the tasks we selected—summarization, helpful response generation, and question-answering—are directly relevant to real-world applications. The datasets used for these tasks are based on real-world Reddit posts, which align closely with practical, user-driven contexts. An example of the task query is “I want to go for a PhD in Philosophy after undergrad: is that stupid?”
>
> By aligning model preferences on these tasks, PROFILE contributes to improving model performance in scenarios that are immediately applicable to practical applications.
>
> ---
>
> **References**\
> [1] Hu, Y., Song, K., Cho, S., Wang, X., Foroosh, H., & Liu, F. (2023). Decipherpref: Analyzing influential factors in human preference judgments via gpt-4. arXiv preprint arXiv:2305.14702. \
> [2] Min, S., Krishna, K., Lyu, X., Lewis, M., Yih, W. T., Koh, P. W., ... & Hajishirzi, H. (2023). Factscore: Fine-grained atomic evaluation of factual precision in long form text generation. arXiv preprint arXiv:2305.14251.

---

> > ### Author Response · Authors · 2024-11-19
> >
> > ### **"does not sufficiently explain the basis for each factor level, the scientific soundness of the classification", "raises concerns about whether the framework can accurately reflect human-model preference differences."**
> >
> > **_(CLARIFICATION)_**  We appreciate the reviewer's request for clarification regarding our factor classification. Our framework is grounded in existing literature, carefully organizing and standardizing previously established factors **(L189-190)**.
> >
> > First, regarding measurement accuracy, we want to clarify that while we present a three-level factor classification as a conceptual framework, our actual measurement of human-model preference alignment is based solely on Level 1 factors.
> >
> > The primary motivation for our classification was to address inconsistent factor definitions and granularity across studies. For example, within the same summarization task,
> > - [1] uses a broad concept of relevance ("to what extent the output is relevant to the source") as an evaluation criterion.
> > - [2] breaks this down into multiple factors like "intent-alignment" and "off-focus".
> > - [3,4] specify it more narrowly by evaluating "whether the summary contains only the important information of the source document."
> >
> > **(Level 1, 2)** To address these variations, we classified existing factors into Level 1 and Level 2 based on semantic similarity and the level of detail captured. For Level 1, we consolidated definitions from [1, 2, 3, 4] for summarization, [5, 6, 7] for helpfulness and QA tasks. Level 2 largely builds on the high-level classification in [2], with additional factors for helpfulness and QA.\
> > **(Level 3)** Moreover, different studies often use identical terms to convey different concepts. For example, "consistency" is defined as "factual alignment between the summary and the source document" in [2, 3], but as "generated text being consistent in the information it provides" in [1]. While the first definition evaluates consistency between the source document and the output, the second focuses solely on the output. To clarify what these factors measure, we introduced a relationship-based classification at Level 3.
> >
> > ---
> >
> > ### **“lack clear methodological and empirical support for guiding improvements in human-AI alignment”**
> > **_(CLARIFICATION, REVISION)_** Our work provides concrete empirical evidence for improving human-AI alignment through PROFILE in both evaluation and generation contexts **(Section 4.4)**. We show that PROFILE can enhance evaluation performance through targeted factor-level guidance. Specifically, by identifying the most misaligned factor and incorporating tailored guidance in evaluator prompts, we achieve better evaluation performance, as shown in **Table 2** (Guide_Mis). The main text **(L406-408)** has been updated to clarify this process, with detailed prompts and procedures outlined in **Appendix F.1.**
> >
> > Our approach to improving factor-level alignment builds on established methods [8, 9] that leverage evaluative feedback to refine generations. While PROFILE primarily focuses on factor-level alignment rather than generation quality, our approach offers practical benefits of fine-grained steerability of model generations. As reviewer [gJUL] noted, our *“findings and insights could be valuable for researchers interested in human alignment, LLMs, and explainable AI.”*
> >
> > ---
> >
> > **References**\
> > [1] Fu, J., Ng, S. K., Jiang, Z., & Liu, P. (2023). Gptscore: Evaluate as you desire. arXiv preprint arXiv:2302.04166.\
> > [2] Hu, Y., Song, K., Cho, S., Wang, X., Foroosh, H., & Liu, F. (2023). Decipherpref: Analyzing influential factors in human preference judgments via gpt-4. arXiv preprint arXiv:2305.14702.\
> > [3] Zhong, M., Liu, Y., Yin, D., Mao, Y., Jiao, Y., Liu, P., ... & Han, J. (2022). Towards a unified multi-dimensional evaluator for text generation. arXiv preprint arXiv:2210.07197.\
> > [3] Fabbri, A. R., Kryściński, W., McCann, B., Xiong, C., Socher, R., & Radev, D. (2021). Summeval: Re-evaluating summarization evaluation. Transactions of the Association for Computational Linguistics, 9, 391-409.\
> > [5] Glaese, A., ... & Irving, G. (2022). Improving alignment of dialogue agents via targeted human judgements. arXiv preprint arXiv:2209.14375.\
> > [6] Ye, S., ... & Seo, M. (2023). Flask: Fine-grained language model evaluation based on alignment skill sets. arXiv preprint arXiv:2307.10928.\
> > [7] Nakano, R., ... & Schulman, J. (2021). Webgpt: Browser-assisted question-answering with human feedback. arXiv preprint arXiv:2112.09332. \
> > [8] Bai, Y., ... & Kaplan, J. (2022). Constitutional ai: Harmlessness from ai feedback. arXiv preprint arXiv:2212.08073.\
> > [9] Xu, W., ... & Wang, W. (2024, August). Pride and prejudice: LLM amplifies self-bias in self-refinement. In Proceedings of the 62nd Annual Meeting of the Association for Computational Linguistics (Volume 1: Long Papers) (pp. 15474-15492).

---

> > > ### Author Response · Authors · 2024-11-19
> > >
> > > ### **“analysis mostly depends on correlation and quantitative scores, lacking qualitative insights into why the model exhibits inconsistencies with human preferences for certain factors.”**
> > >
> > > **_(MORE ANALYSIS, REVISION)_**
> > > We appreciate the reviewer’s suggestion to include more qualitative analysis. In response to the feedback, we've added several examples in **Appendix B** to better illustrate the divergence between human and model preferences.
> > >
> > > Our analysis shows that while models almost always generate longer responses with more atomic facts as higher-scoring outputs **(Table 5)**, examples of model evaluations show that they can still correctly select summaries preferred by human annotators, even if those summaries contain fewer facts or are shorter.  This again highlights model behavior differences between generation and evaluation.
> > >
> > > Additionally, we examined cases where models assigned higher scores to outputs that were *not* favored by human annotators (based on majority votes). Misalignment was more evident when there were larger score gaps. In such cases, humans typically preferred:
> > > - Shorter, less dense summaries
> > > - More informal language (e.g., "guy" vs. "18-year-old male")
> > > - Casual phrasing (e.g., "gotten back together" vs. "renewed romantic connection")
> > >
> > > This aligns with our factor-level findings, where human-preferred outputs, particularly on Reddit-based content, leaned towards lower formality and information, as long as the main idea of the post was included.
> > >
> > > ---
> > >
> > > ### **Q. I am curious about the correlations between different factors. Could you provide an analysis on this?**
> > >
> > > **_(MORE ANALYSIS, REVISION)_** Thank you for suggesting an analysis of the correlations between factors. We conducted a correlation analysis on the distribution of feature scores for each factor in the samples generated by three models (GPT-4o, Gemini-1.5-Pro, and Tulu-2.5-PPO-13B) across our three tasks. We report the results **(Figure 5)** and added detailed analysis in **Appendix G.3**. Below is a summary of our findings:
> > >
> > > 1. Summarization (Reddit TL;DR)
> > > - The tendency of feature correlation was similar in the three models.
> > > - Representatively, there is a high correlation between {*# facts* and *length*} and {*# facts* and *source coverage*}. It is intuitive that the larger the number of facts an answer contains, the longer the answer, and the higher the likelihood of containing more information from the source.
> > >
> > > 2. Helpfulness (SHP-2)
> > > - All three models consistently show a high correlation between every pair among {*length*, *# facts*, *helpfulness*}. This is expected, as longer responses are more likely to include helpful advice or detailed information, in addition to the explanation on {*length* and *# facts*} above.
> > > - In GPT-4o, there is a noticeable trend of high correlation between *receptiveness* and {*helpfulness*, *# facts*, *coherence*, *length*}. As highlighted in **Table 6 (Appendix F)**, these are the four factors GPT-4o tends to prioritize during generation.
> > > - The strong correlations likely result from GPT-4o’s tendency to generate outputs emphasizing these features, suggesting that it considers these factors more frequently during response generation.

---

> > > > ### Comment · Reviewer_ZMM8 · 2024-11-20
> > > >
> > > > Thank you for your thoughtful updates. Most of my concerns have been addressed. I have increased the overall score from 3 to 5.

---

> ### Author Response · Authors · 2024-11-20
> **Have you read our comments?**
>
> Dear Reviewer,
>
> Thank you. We are glad our updates and clarifications were helpful. If you have any further questions, please feel free to let us know.
>
> Best, Authors

---

### Official Review · Reviewer_gJUL · 2024-11-06

**Soundness:** 2
**Presentation:** 2
**Contribution:** 2
**Rating:** 5
**Confidence:** 4

**Summary:**

This paper studies the preference discrepancy between human judgment and model judgment. Specifically, it investigates three RQs using a unified framework, PROFILE, to understand and enhance preference alignment at a fine-grained level (length, hallucination, etc). The authors conduct experiments on three datasets across two settings, and the results suggest misalignment in the generation setting. Their analysis highlights the potential of the proposed model to further improve alignment.

**Strengths:**

The idea of `enhancing' alignment by better understanding the preference discrepancy between human and model judgment through more fine-grained factors is both interesting and important.


Their findings and insights could be valuable for researchers interested in human alignment, LLMs, and explainable AI.


I liked the overall organization of the paper, which consistent with the state of the field.

**Weaknesses:**

The paper would benefit from more precise notation to improve clarity. I found the notation to be inconsistent and, at times, confusing, which impacts readability. For example:

- L134-135: The notation of score level s and Score(r) is confusing. If Score(r) already equals s, what is the purpose of having the model assign a score again?

The writing quality could also be improved. The core value of this paper lies in the exploration of the preference discrepancy between human and model judgment, yet the novelty and key ideas are not clearly articulated. Some concepts and terms are introduced without sufficient explanation, leading to confusion. For example:

- L40-41: “considering their alignment not only as generators but also as evaluators becomes crucial”

A better discussion and analysis are needed. Some findings and conclusions lack depth and specificity. For example:

- L471: "… engage in reward hacking by generating overly lengthy outputs…"  it is unclear how these conclusions were reached. The proposed method does not appear to involve RLHF/DPO training (correct me if I’m wrong).

The so-called "generalizability" conclusion seems to be derived solely from the summarization experiment, which may not provide sufficient support.”

**Questions:**

What factors should we consider regarding preference discrepancies in other tasks, such as math and coding?

Aside from reward models, how do you think RLHF/DPO contributes to alignment?

---

> ### Author Response · Authors · 2024-11-22
>
> We thank the reviewer for recognizing that our work is **"both interesting and important."** We appreciate their view that our findings could be **"valuable for researchers interested in human alignment, LLMs, and explainable AI"** and their positive assessment of our paper's organization. In response to their feedback about writing clarity, we have made revisions throughout the draft, marked in red. Below, we will address the reviewer's specific comments and questions in detail.
>
> ---
>
> ### **“would benefit from more precise notation to improve clarity.”, “L134-135: The notation of score level s and Score(r) is confusing. If Score(r) already equals s, what is the purpose of having the model assign a score again?”**
>
> **_(CLARIFICATION, REVISION)_** Thank you for highlighting areas in the explanation that could be improved. We have updated **Sections 2 and 3** to clarify our definitions and methodology.
>
> To address this, we would like to clarify the scoring process:
>
> We operationally define the preferences of a model through the responses it generates at predefined score levels. Specifically, we prompt the model to generate responses that match a set of predefined scores, represented as R = {r_score | score ∈ {1, 2, 3, 4, 5}}. This systematic prompting ensures that the model's preferences are derived based on its own scoring mechanism. Please refer to the prompt example in **Appendix E.1.1** for additional clarity.
>
> We hope this resolves the concern and aligns with your expectations for the explanation.
>
> ---
>
> ### **“The writing quality could also be improved." "... the novelty and key ideas are not clearly articulated.”**
>
> **_(REVISION)_** We appreciate your valuable feedback on the clarity of our writing. Your comments have been instrumental in helping us significantly improve our manuscript. We have made the following revisions:
>
> - **Introduction**: Refined the organization and provided a clearer articulation of our methodology's novelty.
> - **Sections 2 and 3**: Improved the clarity of operational definitions and notation.
>
> ---
>
> ### **“A better discussion and analysis are needed. Some findings and conclusions lack depth and specificity.”**
>
> **_(REVISION)_** Upon review, we realized that the phrasing in the relevant paragraph may have been misleading. To address this, we have revised the discussion in **Lines 474–482** to enhance clarity and provide more depth and specificity to the findings and conclusions.
>
> Please let us know if further refinements are needed.
>
> ---
>
> ### **"The so-called "generalizability" conclusion seems to be derived solely from the summarization experiment, which may not provide sufficient support."**
>
> We acknowledge that the term "generalizability" has been used in multiple contexts throughout the paper, which may have caused confusion. Below, we clarify the distinct uses of "generalizability" and the revisions made to address potential misunderstandings:
>
> 1. Generalizability of our experiments with score-based generation
>     - **_(CLARIFICATION)_**: In **the third paragraph of Section 5**, we analyzed whether our score-based generation approach—where models generate responses on a 1–5 scoring scale—produces findings that generalize to standard, unconstrained model responses. Specifically, we examined which score-level responses most closely resemble standard responses without predefined scores.
>
>     - **_(REVISION)_**: To better reflect the focus of this discussion, we have renamed the paragraph from "Generalizability of Our Results" to "Validity of the Score-Based Generation Approach." Additionally, we expanded our analysis beyond summarization to include other tasks, providing stronger empirical evidence to support our conclusions. Please see **Table 10 in the Appendix H.**
>
> 2. Generalizability of the framework
>     - **_(CLARIFICATION)_** We want to clarify that PROFILE is designed as a generalizable framework rather than a task-specific methodology. We intentionally chose three representative tasks—summarization, helpfulness, and document QA—because the factors in these tasks encompass core aspects of preference alignment tasks, making our findings broadly relevant to similar tasks. We have conducted experiments across all three tasks, with results beyond summarization (Helpfulness and QA) detailed in **Appendix Table 7-9**. We revised the parts where the writing might misleadingly refer to the generalizability of the results rather than the generalizability of the framework.

---

> > ### Author Response · Authors · 2024-11-22
> >
> > ### **Q1. "What factors should we consider regarding preference discrepancies in other tasks, such as math and coding?"**
> >
> > We intentionally chose not to include tasks like coding and mathematics in our evaluation, as these domains are predominantly assessed based on output "correctness," which lies outside the primary focus of our study on preference alignment. While correctness is primary for Math and Coding tasks, [1] outlines additional important factors: solution conciseness and completeness in Math, and code efficiency and readability in programming. These preference factors could be analyzed using our framework.
> >
> > ---
> >
> > ### **Q2. "Aside from reward models, how do you think RLHF/DPO contributes to alignment?"**
> >
> > While our analysis is limited to the Tulu model, our experimental findings within the same-sized Tulu-13B model provide some interesting insights:
> >
> > - The Tulu-13B + DPO model demonstrated higher alignment compared to the Tulu-13B + PPO model in terms of factor-level preference across both generation and evaluation settings **(Table 9 in Appendix G.2)**.
> > - When comparing the preference alignment of the Tulu-13B + PPO model and the reward model, the reward model exhibited the highest alignment, followed by the PPO model in the evaluation setting, and finally, the PPO model in the generation setting **(Figure 4)**.
> >
> > We hypothesize that models directly learning preferences (Reward Models and DPO-trained models) may experience less information loss when trained on human preference datasets,  compared to PPO models which rely on an intermediary Reward Model to learn preferences.
> >
> > However, we emphasize that these observations are preliminary and based solely on data from the Tulu model. Other hypotheses and influencing factors may explain these results, and further research is needed to confirm and generalize these findings.
> >
> > ---
> >
> > **References**
> >
> > [1] Ye, S., Kim, D., Kim, S., Hwang, H., Kim, S., Jo, Y., ... & Seo, M. (2023). Flask: Fine-grained language model evaluation based on alignment skill sets. arXiv preprint arXiv:2307.10928.

---

> > > ### Author Response · Authors · 2024-11-25
> > >
> > > Hi Reviewer gJUL,
> > >
> > > Thank you again for your time and efforts on our work. Your feedback has been invaluable in revising our paper. We have carefully addressed each of your concerns. If you have any additional questions, we would be happy to discuss them further during this open discussion period. If you find no further concerns, we kindly ask you to consider raising the score.  Thank you once again for your time and efforts!
> > >
> > > Best,\
> > > Authors

---

> > > > ### Comment · Reviewer_gJUL · 2024-11-26
> > > >
> > > > Thank you for your detailed response and the updated paper. I appreciate the clarifications and improvements. I will maintain the current scores, as I believe they adequately reflect the quality and contribution of your work.

---

> > > > > ### Author Response · Authors · 2024-11-27
> > > > >
> > > > > Dear Reviewer,
> > > > >
> > > > > Thank you. We are glad our updates and clarifications were helpful. If you have any further questions, please feel free to let us know.
> > > > >
> > > > > Best, Authors

---

### Author Response · Authors · 2024-11-23
**General Response to All Reviewers**

We sincerely appreciate the meaningful and constructive feedback from all reviewers. The insights and questions provided by the reviewers were incredibly helpful in improving our paper during the review process.

We are especially grateful to the reviewers for recognizing our framework's **explainability** (Reviewer vTJd) and **interpretability** (Reviewer ZMM8), as well as for finding it both **interesting and important** (Reviewer gJUL). Furthermore, we are pleased that our findings and insights are considered **valuable for researchers interested in human alignment** (Reviewer gJUL) and are seen as offering **practical implications for developing more human-aligned LLMs** (Reviewer vTJd). We sincerely hope that our contributions will help advance the alignment of LLMs with human preferences.

We have addressed the majority of the reviewers' concerns and would like to summarize our main contributions and the key points addressed as part of our general response.
## **Main Contribution**
- We present **PROFILE**, a framework for analyzing factor-level preferences in human-LLM alignment. Using PROFILE, it is possible to analyze **why** humans and models prefer certain responses **across various tasks** and identify **which factors** contribute to misalignment between humans and models.
- Using PROFILE, we applied our framework to analyze human-model alignment across three commonly used alignment tasks: summarization, helpful response generation, and document-based QA. Our analysis revealed that in all three tasks, models tend to prioritize "length" as a key factor in their generation output, whereas human preferences vary depending on the task.
- The explainable analyses enabled by PROFILE can facilitate **improvements in human-LLM alignment**. For example, our experiments illustrate that misalignments can be improved by providing direct instructions to models about the factors on which they are misaligned. Additionally, these insights could be used as training signals to improve alignment further.

## **Summary of the Discussion Phase and Draft Revision**
We have addressed the majority of the concerns raised by the reviewers and have revised the paper accordingly. These revisions have been highlighted in red for easy identification.

- **Clarification**: We acknowledge that certain parts of the paper, particularly **Section 2: Problem Definition** and **Section 3: Profile**, may have caused confusion, as raised by Reviewer gJUL. To address this, we have revised the ambiguous explanations and made significant adjustments to the writing in Sections 2 and 3 to enhance clarity and improve reader comprehension.

- **New Analysis**:
In response to Reviewer ZMM8's inquiry about the **correlation between factors**, we have conducted and included a detailed analysis (Appendix G.3). We also included a **qualitative analysis** of the models' factor level preferences, including the examples (Table 5 in Appendix). To address Reviewer gJUL's concerns regarding the generalizability of one of our experiments, We have extended our evaluation by incorporating an **analysis of two additional tasks alongside summarization**, as presented in Table 10 of the Appendix.

Lastly, thank you to the reviewers for their invaluable feedback and thoughtful consideration.

---

### Meta-Review · Area_Chair_oiy9 · 2024-12-21

**Metareview:**

This paper explores the differences between human and model judgments, focusing on preference discrepancies. It addresses three research questions using a unified framework, PROFILE, to examine and enhance preference alignment at a granular level, including aspects such as length and hallucination. The authors conduct experiments on three datasets across two settings, revealing misalignment in the generation setting. Their analysis underscores the potential of the proposed model to further improve alignment.

Pros:
1. The idea is both interesting and important. The paper is easy to follow.
2. The findings and insights is useful for researchers interested in human alignment, LLMs, and explainable AI.

Cons:
1. The presentation quality needs further improvement. Some notations are inconsistent and confusing, reducing the readability. The novelty and key ideas are not clearly articulated.
2. The findings and conclusions lack depth and specificity. A better discussion and analysis are needed.
3. Lack sufficient evaluation. The paper might not thoroughly compare with existing methods, leaving questions about its relative advantages. The paper's analysis mostly depends on correlation and quantitative scores, lacking qualitative insights into why the model exhibits inconsistencies with human preferences for certain factors. This approach results in a somewhat superficial view that fails to reveal the deeper reasons behind the observed divergences.
4. There are concerns about the scalability of the proposed framework without fine-grained human annotations, impacting its practicality.
5. Potential issues in the approach. The study uses GPT-4o for factor quantification while examining human-AI preference alignment, which may introduce new biases, potentially affecting the objectivity of the analysis.

I agree with most of the reviewers that this paper would benefit from another round of major revision.

**Additional Comments On Reviewer Discussion:**

While the authors address some of the questions during the rebuttal, there are still major concerns remain, as listed in the Cons, and thus the paper is not ready for publishing.

---

### Decision · Program_Chairs · 2025-01-22

Reject